# Tropical deforestation causes large reductions in observed precipitation

C. Smith[1✉], J. C. A. Baker[1] & D. V. Spracklen[1]

Tropical forests play a critical role in the hydrological cycle and can influence local and regional precipitation[1]. Previous work has assessed the impacts of tropical deforestation on precipitation, but these efforts have been largely limited to case studies[2]. A wider analysis of interactions between deforestation and precipitation— and especially how any such interactions might vary across spatial scales—is lacking. Here we show reduced precipitation over deforested regions across the tropics. Our results arise from a pan-tropical assessment of the impacts of 2003–2017 forest loss on precipitation using satellite, station-based and reanalysis datasets. The effect of deforestation on precipitation increased at larger scales, with satellite datasets showing that forest loss caused robust reductions in precipitation at scales greater than 50 km. The greatest declines in precipitation occurred at 200 km, the largest scale we explored, for which 1 percentage point of forest loss reduced precipitation by 0.25 ± 0.1 mm per month. Reanalysis and station-based products disagree on the direction of precipitation responses to forest loss, which we attribute to sparse in situ tropical measurements. We estimate that future deforestation in the Congo will reduce local precipitation by 8–10% in 2100. Our findings provide a compelling argument for tropical forest conservation to support regional climate resilience.

Tropical forests play an important role in moderating local, regional and global climate through their impact on energy, water and carbon cycles[3]. Crucially, tropical forests control local-to-regional rainfall patterns[1,2]. Evapotranspiration from tropical forests is a strong driver of regional precipitation[4,5] contributing up to 41% of basin mean rainfall over the Amazon and up to 50% over the Congo[6]. Evergreen tropical forests are dependent on high annual rainfall for their survival and productivity[7], and forest–rainfall feedbacks have been highlighted as an important determinant of tropical forest stability[4,5,8], amid concerns that the exacerbating impacts of droughts and deforestation could threaten their viability[9].

Rapid loss of forests is occurring across the tropics[10]. Tropical deforestation warms the climate at local-to-global scales by changing the surface energy balance and through emissions of carbon dioxide[3]. The impact of tropical deforestation on precipitation is less certain with a range of processes operating at different scales. Small-scale deforestation over the southern Amazon has been shown to increase precipitation frequency[11,12] owing to thermally[13] and dynamically[12] induced circulations. At larger scales, deforestation reduces precipitation recycling leading to a reduction in precipitation[1,14]. Over Indonesia, deforestation has been linked to declining precipitation[15], and exacerbation of El Niño impacts[16]. Global and regional climate models predict annual precipitation declines of 8.1 ± 1.4% for large-scale Amazonian deforestation by 2050 (ref. [17]), but an observational study of the impacts of tropical deforestation on precipitation across spatial scales is lacking.

Here we present a pan-tropical assessment of the impact of forest loss on precipitation based on measurements. We use a satellite dataset of forest cover change over the period 2003–2017 to identify areas of forest loss, with a focus on evergreen broadleaf forests of the Amazon, Congo and Southeast Asia (SEA; Fig. 1). To provide a robust assessment of the impacts of deforestation on precipitation, we analysed 18 different precipitation datasets, including satellite ($n = 10$), station-based ($n = 4$) and reanalysis ($n = 4$) products (Extended Data Table 1). We compared the precipitation change over pixels experiencing forest loss with neighbouring pixels that had experienced less forest loss (Methods). This comparison against neighbouring pixels that will have experienced similar climate change focuses our analysis on the changes due to forest loss. To explore the impact of forest loss across scales, we analysed the impacts of forest loss on coincident precipitation at a series of spatial resolutions ranging from roughly 5 km to 200 km (0.05°, 0.1°, 0.25°, 0.5°, 1.0° and 2.0°).

## Precipitation response to forest loss

Observed precipitation responses to tropical forest loss across multiple spatial scales and precipitation products are presented in Fig. 2. Satellite-based precipitation datasets suggest that tropical forest loss causes statistically significant ($P < 0.05$) declines in median annual mean precipitation at all scales analysed. At larger scales (>0.5°), reductions exceed 0.03 mm per month for each percentage point loss of forest cover (Fig. 2d–f). The largest changes are observed at the 2.0° scale (approximately 220 km at the Equator; Fig. 2f), for which each percentage point reduction in forest cover causes 0.25 ± 0.1 mm per month reduction in annual precipitation.

Analysis of precipitation change as a function of forest loss confirms larger reductions in precipitation for larger reductions in forest cover

[1]School of Earth and Environment, University of Leeds, Leeds, UK. ✉e-mail: ee13c2s@leeds.ac.uk

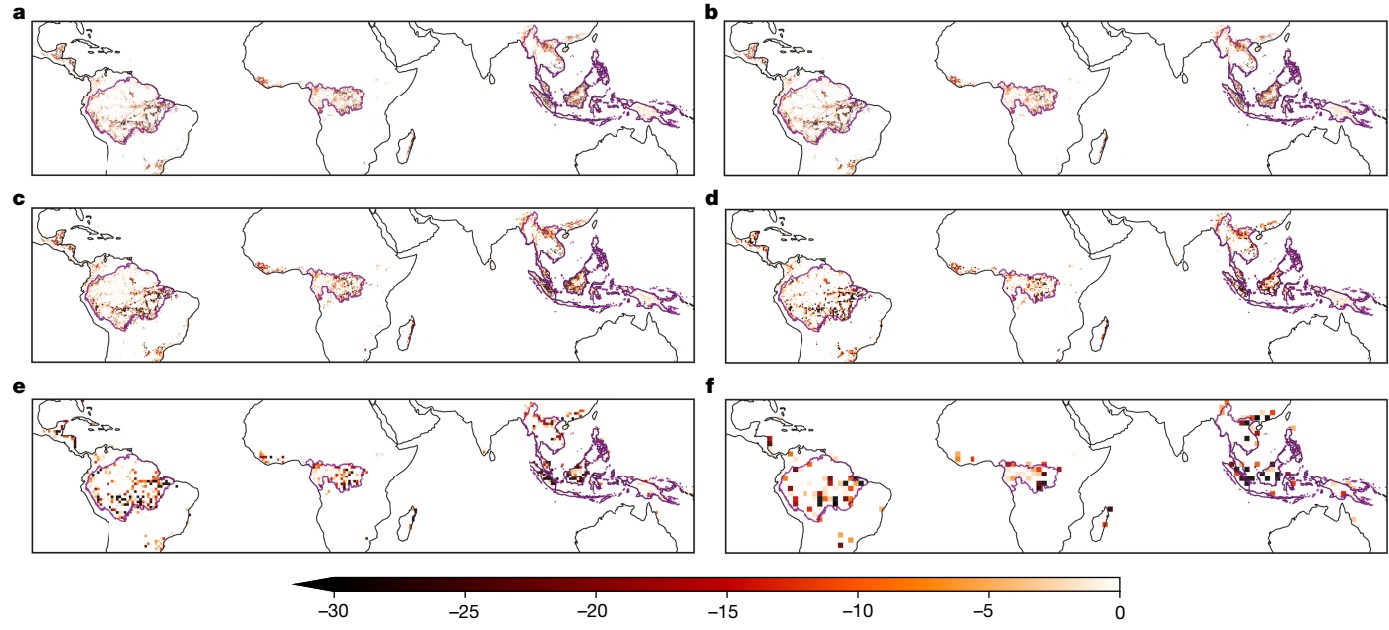

**Fig. 1 | Tropical evergreen broadleaf forest cover loss from 2003 to 2017.**
**a**–**f**, Forest cover change at 0.05° (**a**), 0.1° (**b**), 0.25° (**c**), 0.5° (**d**), 1.0° (**e**) and 2.0° (**f**) resolution. The Amazon Basin, Congo Basin and SEA regions used in this study are outlined in purple. Maps of the different regions generated using Cartopy and Natural Earth[51]. Forest loss data from ref.[10].

(Extended Data Fig. 1), although with considerable variability, as seen in the modelled response[18]. Observed reductions in precipitation are consistent across satellite datasets, with all ten satellite precipitation products agreeing on the sign of the rainfall response at 2° over the tropics (Extended Data Fig. 2). At the 2° scale, significant ($P < 0.05$) reductions in annual mean precipitation with forest loss were observed across all tropical regions (Fig. 2). Reductions in precipitation at 2° based on satellite datasets ranged from 0.48 ± 0.36 mm per month in

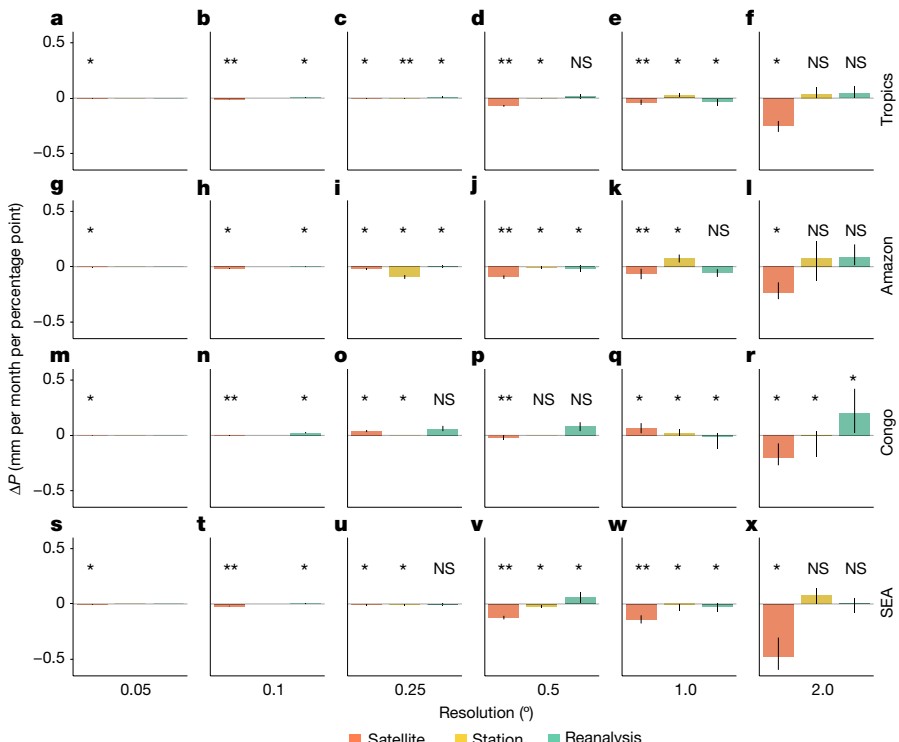

**Fig. 2 | Reductions in precipitation over regions of tropical forest loss.**
**a**–**r**, Bars indicate the median absolute change in annual precipitation (millimetres per month) per percentage point of forest loss over 2003 to 2017 in each region (tropics (**a**–**f**), Amazon (**g**–**l**), Congo (**m**–**r**), SEA (**s**–**x**)) for each precipitation dataset category (satellite, station and reanalysis). Results are shown for forest loss scales of 0.05° (**a**,**g**,**m**,**s**), 0.1° (**b**,**h**,**n**,**t**), 0.25° (**c**,**i**,**o**,**u**), 0.5° (**d**,**j**,**p**,**v**), 1.0° (**e**,**k**,**q**,**w**) and 2.0° (**f**,**l**,**r**,**x**). Statistically significant (*$P < 0.05$; **$P < 0.01$) and nonsignificant (NS) differences in changes in mean precipitation (calculated as a multi-annual mean over 2003–2007 compared with 2013–2017) over deforested regions compared with control regions are indicated. Error bars show ±1 standard error from the mean. Datasets used in this analysis are detailed in Extended Data Table 1. $\Delta P$, precipitation change.

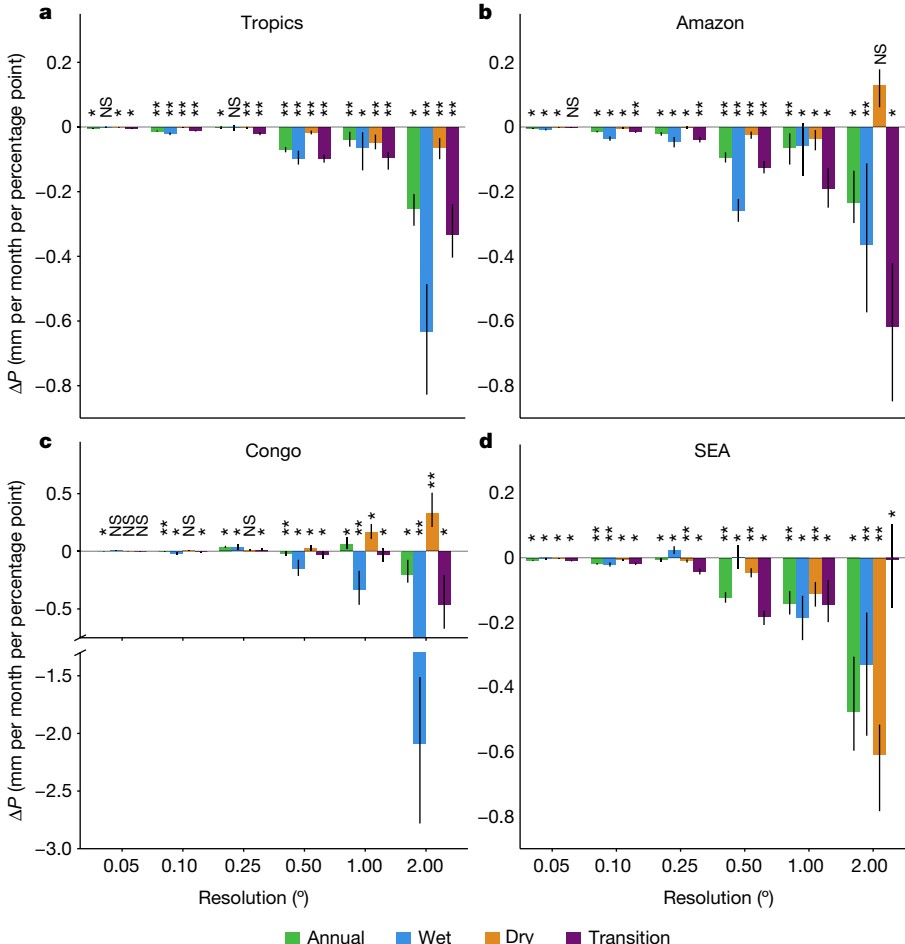

**Fig. 3 | Changes in seasonal precipitation due to forest loss. a–d,** Bars indicate the median change in precipitation (millimetres per month) per percentage point forest cover loss for satellite datasets during 2003–2017 for tropics (**a**), Amazon (**b**), Congo (**c**) and SEA (**d**). Error bars indicate ±1 standard error from the mean. Statistically significant (*$P < 0.05$; **$P < 0.01$) and nonsignificant (NS) differences in changes in mean precipitation over deforested regions compared with controls are indicated. Results are shown for the wettest 3 months (wet), the driest 3 months (dry) and the transition months (remaining 6 months). Datasets used in this analysis detailed in Extended Data Table 1.

SEA to $0.23 \pm 0.12$ mm per month in the Amazon, and $0.21 \pm 0.19$ mm per month in the Congo for each percentage point loss in forest cover, with at least 8 out of 10 satellite datasets agreeing on the sign of the response within each region (Extended Data Fig. 2). In SEA, it has been suggested that proximity to the ocean and the replacement of tropical forest with plantations as opposed to pasture or cropland may cause reduced sensitivity of precipitation to deforestation[1]. Our analysis suggests that forest loss in SEA causes reductions in precipitation consistent with or greater than reductions in precipitation in the Amazon and Congo.

Station-based datasets and reanalysis products exhibit contrasting annual mean precipitation responses to deforestation at 2.0° (Fig. 2). Across the tropics, station-based and reanalysis datasets showed no statistically significant changes in annual mean precipitation due to forest loss (Fig. 2f), and there was little agreement with satellite datasets at the regional scale (Fig. 2l,r,x), with some non-satellite precipitation products showing small increases in annual mean precipitation due to forest loss. Sparse in situ measurements across the tropics, particularly over regions of forest loss, mean that station-based datasets provide a weak constraint on precipitation changes. A comparison of station-based precipitation datasets revealed higher levels of uncertainty in the tropics, including the Amazon[19]. In regions of sparse data such as tropical forests[20], interpolation methods may mask precipitation changes driven by forest loss. Reanalysis products, which are numerical models constrained by empirical data, are also expected to

be less reliable in regions where in situ data are limited[21]. Our results indicate that precipitation data based on satellite remote-sensing measurements may have an advantage over tropical forest regions where in situ measurements are sparse or unavailable. For these reasons, we focus our analysis on satellite-based datasets and identify where agreement between datasets exists.

Our results are robust (Extended Data Fig. 3) to a range of methodological assumptions including the length of analysis period, the choice of start and end period and the spatial extent of control pixels (Methods). Our analysis period includes the 2015–2016 El Niño that resulted in negative precipitation anomalies over many tropical land regions (Supplementary Fig. 1). We found that the precipitation response to forest loss was robustly negative during both El Niño and non-El Niño years (Extended Data Fig. 3). Over the Amazon and SEA, we see a stronger reduction in precipitation over regions of forest loss during El Niño years. The relative impact of El Niño on precipitation is smaller in the Congo[22], and correspondingly we do not see a stronger reduction here. A stronger precipitation response to forest loss in regions and periods impacted by El Niño is probably due to higher transpiration rates observed in tropical forests during El Niño years[23] and because rainfall is more sensitive to reductions in moisture recycling during drought years[5,24]. Climate change is expected to lead to increased droughts over many tropical regions[25], which may be further exacerbated by ongoing deforestation.

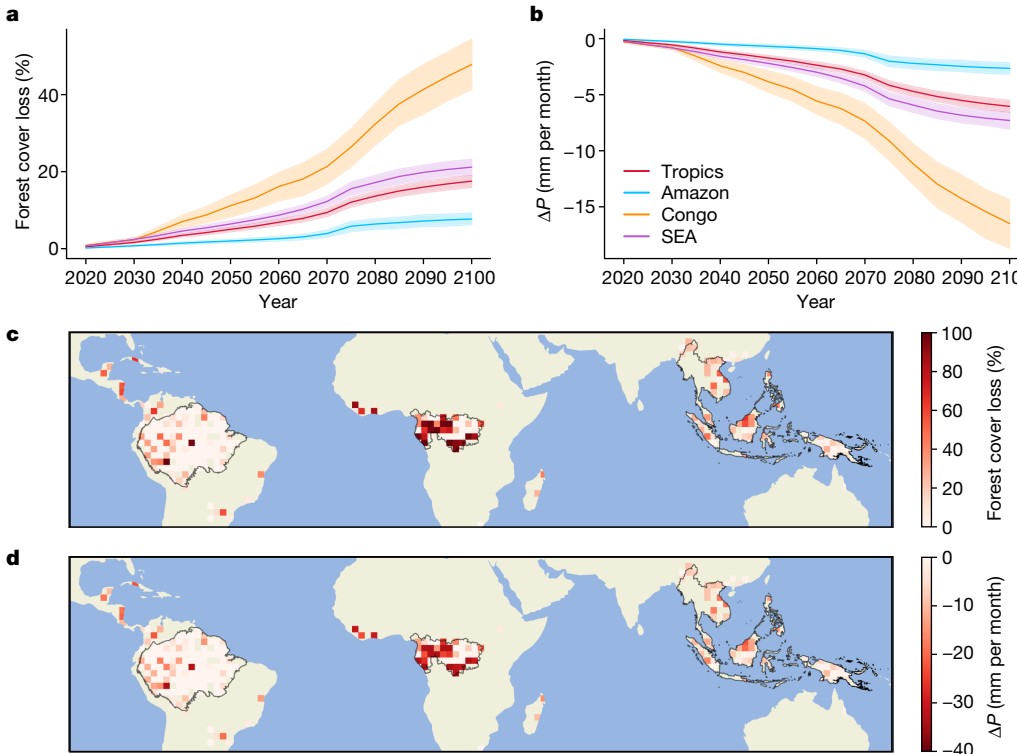

**Fig. 4 | Impact of projected future forest loss on annual mean precipitation.** **a**, Mean forest cover loss over 2015–2100 under Shared Socioeconomic Pathway 3–Representative Concentration Pathway 4.5 for the tropics, Amazon, Congo and SEA. **b**, Impact of projected forest cover loss on precipitation ($P$; ±1 standard error from the mean). **c**, Spatial pattern of forest cover loss. **d**, Predicted $P$ change ($\Delta P$) in 2100 due to forest cover loss. Results are shown for 2.0° resolution. Maps of the different regions generated using Cartopy and Natural Earth[51].

## Seasonal precipitation reductions

Changes in precipitation due to forest loss during the dry, wet and transition seasons are nearly consistently negative (Fig. 3). For the tropics, absolute changes in precipitation with forest loss are greatest in the wet season (Fig. 3a, up to −0.6 mm per month per percentage point forest loss) whereas relative changes of precipitation with forest loss are similar (−0.2% per percentage point) across dry, wet and transition seasons (Supplementary Fig. 2). In the Amazon, deforestation causes the largest reductions in precipitation during the transition season (Fig. 3b) as has been found previously[18,26,27].

Previous case studies have indicated that dry-season precipitation can increase over deforestation in the Amazon[11,28,29]. We observed a nonsignificant increase in dry-season precipitation due to forest loss in the Amazon at 2° as well as increases in the Congo at 1° and 2° (Fig. 3). In SEA, forest loss causes reductions in dry-season precipitation across all scales (Fig. 3d). The mechanism through which forest loss impacts precipitation is likely to change with both season and spatial scale. At the smallest scales (5 km), thermally driven impacts are likely to dominate, shifting to dynamically driven impacts through reductions to surface roughness, then to reductions in moisture fluxes and precipitation recycling at the largest scales[12,30]. Our observation of greater reductions in precipitation due to deforestation at larger spatial scales is consistent with a reduction in moisture recycling emerging as the dominant mechanism[1].

## Comparison with climate models

A meta-analysis of climate model studies (predominantly global models with >2° resolution) found that forest loss in the Amazon resulted in a mean reduction in annual mean precipitation of 0.16 ± 0.13% per percentage point[17], overlapping with our value of 0.25% per percentage point (Supplementary Fig. 2). Fewer simulations have been conducted for the Congo, with models predicting a reduction in precipitation of 0.16 ± 0.17% per percentage point[2], similar to our reduction of 0.15% per percentage point (Supplementary Fig. 2). The large range of model estimates highlights the substantial uncertainty in model predictions. Our observationally derived analysis provides support for models that predict reductions in precipitation under regional deforestation at global climate model scales.

Our observational analysis documents the impacts of deforestation on precipitation across the tropics. Applying linear scaling to the reductions in precipitation observed in our analysis would suggest that complete deforestation could result in reductions in annual precipitation of 10–20%. Previous estimates of the impact of complete deforestation on precipitation range from a 16% (ref. [17]) to 55–70% (ref. [31]) reduction in the Amazon and an 18% (ref. [2]) to 50% (ref. [32]) reduction in the Congo.

## Impacts of future deforestation

To further explore how future deforestation might modify precipitation, we combined our observationally derived estimates of precipitation responses to forest cover loss with future projections of land cover change from a high-deforestation scenario (Methods). We estimate that forest loss from 2015 to 2100 (Fig. 4a) could lead to reductions of annual mean precipitation of up to 16.5 ± 6.2 mm per month in the Congo (Fig. 4b), equivalent to precipitation declines of 8–10%. Forest loss is projected to be greatest in the western and southern Congo (Fig. 4c), which will also experience the strongest reductions in precipitation (Fig. 4d).

The sensitivity of precipitation to the extent of forest loss is an uncertainty in our analysis, a result of the relatively short observational record, compounded by large spatial and temporal variability in precipitation. The response of precipitation to forest loss greater than

30%, a threshold beyond which large reductions in precipitation have been postulated[1], is one such uncertainty. Restricting our analysis to the forest losses of 0–30% that are well sampled in our observational dataset (Supplementary Fig. 3), through capping the impacts of greater forest loss at that of 30%, results in projected annual mean precipitation reductions of 6.5 ± 2.6 mm per month in the Congo and 6.2 ± 2.5 mm per month in SEA (Supplementary Fig. 4). However, restricting our analysis in this way is likely to underestimate the precipitation impacts over regions projected to experience the most extensive deforestation, including the Congo, where mean forest cover is projected to decline by 40 percentage points between 2015 and 2100 (Fig. 4a).

Previous studies have identified both linear[9,33] and nonlinear[1,31] responses of precipitation to forest loss. Such nonlinear interactions and feedbacks have the potential to further amplify or moderate the responses predicted here[14,34]. Our analysis shows large reductions in precipitation for relatively small amounts of forest loss and evidence for reduced sensitivity of precipitation to additional amounts of forest loss (Extended Data Fig. 1). Assuming a nonlinear relationship between forest loss and precipitation (Methods) reduces our projected reductions in precipitation by around a factor of 2 (Supplementary Fig. 5). Our observationally based approach will miss tipping points in the climate system that might be reached as deforestation extent progresses further[1]. Such tipping points have been postulated for the Amazon under future global change[25,35]. Thus, the substantial declines in precipitation projected in our analysis should be viewed as a conservative estimate of the potential precipitation response to future deforestation. Nevertheless, our analysis suggests that deforestation can drive local and regional precipitation changes that may match or exceed those predicted due to climate change over the same period[36,37].

## Implications of precipitation reductions

Reductions in precipitation induced by forest loss have important implications for society and the sustainability of remaining tropical forest. Deforestation-induced reductions in precipitation affect agriculture[1,14] and hydropower generation[38]. On average, crop yields decline by 0.5% for each percentage point reduction in precipitation[39]. Our results indicate that forest-loss-induced changes to annual precipitation (Supplementary Fig. 2) could cause crop yields to decline by 1.25% for each 10-percentage-point loss of forest cover, potentially exacerbating the impacts of climate change and future drought events. The maintenance of regional rainfall patterns due to forests in the Amazon has been valued at up to US$9 ha$^{-1}$ yr$^{-1}$ and US$1.84 ha$^{-1}$ yr$^{-1}$ through sustaining agricultural yields and hydropower generation, respectively[40]. Global cropland area increased by 9% in the past two decades, with even higher increases in South America and tropical Africa[41] largely at the expense of natural ecosystems. Further agricultural expansion in tropical forest regions may lead to overall reductions in production if declines in yield due to deforestation-induced reductions in rainfall outweigh increased production from expanded agricultural area[14].

Furthermore, reductions in rainfall over remaining areas of tropical forest are expected to lead to additional forest loss[9] as well as impacting species composition[22], carbon sequestration[42] and fire frequency[43]. Reductions in dry-season precipitation pose a particular threat to forest viability by exacerbating seasonal droughts and potentially delaying the onset of the wet season and extending the length of the dry season. Increases in dry-season length over recent decades have previously been reported for the Amazon[44] and the Congo[45], possibly linked to land cover changes[27].

Deforestation may also shift precipitation patterns, increasing dry-season rainfall immediately downwind of forest loss and decreasing rainfall in upwind areas[12]. Our approach is restricted to observing deforestation impacts up to scales of 200 km (Methods). At larger scales, insufficient pixels experienced forest loss during the relatively short period of satellite observations for a robust analysis. Deforestation is also likely to alter precipitation at these larger scales through reducing moisture recycling leading to reductions in rainfall downwind of forest loss[4,5,9,35]. The length scale of moisture recycling has been estimated at 500–2,000 km in the tropics[46], with a median value of 600 km in the Amazon[5]. In regions downwind of extensive forests, such as the southwestern Amazon, up to 70% of precipitation could be sourced from upwind evapotranspiration[47,48]. Tropical forest loss could therefore have severe implications for precipitation in these regions that are hundreds to thousands of kilometres downwind of the forest loss[5]. Through missing the impacts at these larger scales, our analysis is likely to underestimate the full impacts of deforestation on rainfall.

Our results highlight the importance of remaining tropical forests for sustaining regional precipitation. Despite efforts to reduce deforestation, rates of tropical forest loss have accelerated over the past two decades[49]. Renewed efforts are needed to ensure recent commitments to reduce deforestation, including the New York Declaration on Forests and The Glasgow Leaders' Declaration on Forests and Land Use made at the 26th UN Climate Change Conference of the Parties, are successful. Global efforts to restore large areas of degraded and deforested land could enhance precipitation[50], reversing some of the reductions in precipitation due to forest loss observed here.

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

## Methods

### Datasets

We used 18 precipitation datasets, listed in Extended Data Table 1. All datasets were downloaded at the highest available spatial resolution, which for some datasets was 0.04°, or approximately 4 km at the Equator. Data were obtained as monthly means or converted to monthly mean using the Python package xarray[52]. We categorized precipitation datasets as satellite ($n = 10$), station ($n = 4$) and reanalysis ($n = 4$). Satellite datasets are those based primarily on data from satellite sensors and include datasets that have both satellite and station-based data (that is, merged datasets). Station datasets include only ground-based information from weather stations and rain gauges. Reanalysis products are models constrained by surface and satellite data. Precipitation datasets have been compared previously over the Amazon[20] highlighting the limited station data over tropical forest regions. Time series of precipitation (Supplementary Fig. 1) reveal variability across the different datasets highlighting the need to analyse impacts of deforestation across multiple datasets.

To analyse the changes in forest canopy cover, we used data from the Global Forest Change (GFC) version 1.9 (ref. [10]). GFC v1.9 provides forest canopy cover in the year 2000 and subsequent annual forest loss from 2001–2020 at 30-m resolution. We analysed forest cover and precipitation changes over the period 2003 to 2017, which was the period common to all datasets.

### Analysis across multiple spatial scales

We analysed the impacts of forest loss across a range of scales (0.05°, 0.1°, 0.25°, 0.5°, 1.0° and 2.0°). Each precipitation dataset was analysed at its native resolution and at all lower resolutions across this range of scales. Spatial regridding was carried out using the Python package xESMF[53] with a bilinear regridding scheme. Two alternative regridding methods (xESMF: conservative-normalized; and iris: area weighted) were tested and had little impact on our results. For GFC data, we calculated forest loss using the original 30-m data and converted the resulting values to each of the six spatial resolutions analysed by taking the sum of all 30-m pixels within each larger pixel. Change in canopy cover from 2003 to 2017 at each resolution is shown in Fig. 1.

### Assessing impact of historical deforestation on precipitation

We used a moving-window nearest-neighbour approach[54] to compare the forest loss and precipitation change of each pixel with that of its immediate neighbours. We tested the sensitivity of the analysis to the size of the moving window and found similar results for 3 × 3 and 5 × 5 (Extended Data Fig. 2) moving windows. Results from the 3 × 3 moving-window approach can been seen in the main paper. We calculated the forest loss of each deforested pixel relative to neighbouring control pixels as the forest loss of the deforested pixel minus forest loss of the control. We constrained our analysis to the tropical evergreen broadleaf biome using the Moderate Resolution Imaging Spectroradiometer land cover dataset[55]. To be included in the analysis, deforested pixels must have experienced 0.1% more forest loss over time than their neighbouring control pixels. The number of deforested pixels analysed varied between analysis resolutions as follows: 0.05°, $n = 243{,}254$; 0.1°, $n = 58{,}660$; 0.25°, $n = 9{,}604$; 0.5°, $n = 2{,}303$; 1.0°, $n = 586$; 2.0°, $n = 123$. We observed similar distributions of canopy change for all spatial resolutions analysed (Supplementary Fig. 6).

We calculated the precipitation change of the deforested pixel relative to the precipitation change of the control pixel ($\Delta P$) as the precipitation change of the deforested pixel over the analysis period (for example, 2003–2017) minus the precipitation change over the control pixel. To reduce the impact of interannual variability in precipitation on our results, we calculated 5-yr means for periods at the start (2003–2007) and end (2013–2017; Extended Data Fig. 5) of the analysis period. We then calculated the change in precipitation as

the difference between the start and end of these multi-year means. We report precipitation changes ($\Delta P$) as a function of forest loss by dividing by the difference in forest loss between deforestation and control pixels (units of millimetres per month per percentage point). We also report precipitation change as the percentage change in precipitation ($\Delta P/P$, in units of per cent) as a function of forest loss (in units of per cent per percentage point).

To ensure that control pixels and deforested pixels experience a similar background climate, we conducted a sensitivity test in which we restricted our analysis to pixels for which the pre-deforestation precipitation across the control and deforested pixels differed by less than 10%. Restricting our analysis in this way had little impact on our results (Supplementary Fig. 7) showing that our nearest-neighbour approach is effective even at the largest scales analysed here.

To explore the role of the analysis period on our results, we compared the results for 5-yr means to those for shorter 3-yr means (2003–2005 versus 2015–2017) and found consistent results (Extended Data Fig. 3). Our analysis period includes the strong 2015/2016 El Niño that resulted in reductions in precipitation over most tropical land regions, particularly in 2015 (Supplementary Fig. 1). To explore the potential impacts of the 2015/2016 El Niño on our analysis, we estimated the impact of forest loss on precipitation using 3-yr (2003–2005 versus 2018–2020) and 5-yr (2003–2007 versus 2016–2020) multi-annual means spanning an extended time period. The 3-yr analysis completely excludes the 2015/2016 ENSO, and the 5-yr analysis excludes 2015, which was the driest year (Extended Data Fig. 3). Two datasets (TRMM and UDEL) were not available after 2017, so they were removed from this sensitivity analysis.

### Statistical analysis

For each category of precipitation data (satellite, station and reanalysis), precipitation change values were grouped together for all deforestation pixels and all control pixels. We found that precipitation changes for deforested pixels and control pixels, and the difference in precipitation change between deforested and control pixels (Extended Data Fig. 4), were normally distributed. Error bars (Figs. 2 and 3) show ±1 standard error from the mean calculated and displayed using the Python package Seaborn[56]. To test whether mean precipitation changes over regions of deforestation were statistically different from changes over the control areas, we used a Student's *t*-test. We also used the Mann–Whitney test to test for significant differences in median precipitation change between control and deforested pixels and found similar results.

### Seasonal analysis

For the satellite datasets alone, in addition to calculating precipitation changes at the annual timescale, we calculated changes for the dry season (driest 3 months of each year), wet season (wettest 3 months of each year) and transition season (remaining 6 months). The driest, wettest and transition months were identified for each pixel using each individual precipitation dataset. For each season and dataset, we calculated the median change in precipitation across all of the pixels within the region of interest (Supplementary Figs. 8–10).

### Predicting future precipitation change due to forest loss

We used projections of forest cover change available at 0.05° from the Global Change Analysis Model (GCAM) for 2015–2100 based on the Shared Socioeconomic Pathway 3–Representative Concentration Pathway 4.5 scenario, which represents a high-deforestation future[57]. GCAM includes the impacts of climate and land use on future forest cover. We summed forest cover from all forest categories and calculated forest cover loss in each year compared to a 2015 baseline. Forest cover loss data were regridded to 2°. We estimated the impact of forest loss on future precipitation at the 2° scale through multiplying the projected percentage point forest loss for each pixel by the

observed median change in precipitation (millimetres per month) per percentage point forest cover loss across the satellite datasets. To estimate the uncertainty in our predictions, we applied an upper and lower limit on the sensitivity of precipitation to forest loss based on the median value ±1 standard error from the mean (see error bars in Fig. 2) and rescaled by forest loss. This provides a range of estimated precipitation impacts of future forest loss. We also tested the impact on our results of capping future forest loss in each pixel at 30%, which is the upper range of forest loss that is well sampled in the observations (Supplementary Fig. 3). For each region, we applied the tropical satellite precipitation response to forest loss (Fig. 2f), meaning that our projected regional precipitation changes are a product of the regional canopy cover change and the median tropical precipitation response. Our approach assumes a linear precipitation response to forest loss, which recent work suggests could provide a conservative estimate of deforestation impacts[31]. We tested the sensitivity of assuming a linear response of precipitation to canopy cover loss. We fitted a nonlinear function to the data presented in Extended Data Fig. 1 through applying the median sensitivity of precipitation to forest cover loss (millimetres per month per percentage point) within each forest cover loss bin. We then scaled by the projected forest cover loss. This approach reduces the projected reduction in precipitation to 2.4 mm per month in SEA and 1.5 mm per month in the Congo (Supplementary Fig. 5).

## Data availability

Full results for all tested resolutions used in this analysis are available through https://doi.org/10.5281/zenodo.7373832. The original datasets are freely available to download from the following repositories: CHIRPS from https://data.chc.ucsb.edu/products/?C=M;O=D, CMORPH from https://ftp.cpc.ncep.noaa.gov/precip/CMORPH_RT/GLOBE/data/, CPC from https://psl.noaa.gov/data/gridded/data.cpc.globalprecip.html, CRU from https://crudata.uea.ac.uk/cru/data/hrg/, ERA5 from https://cds.climate.copernicus.eu/cdsapp#!/dataset/reanalysis-era5-single-levels?tab=overview, GPCC from https://opendata.dwd.de/climate_environment/GPCC/html/download_gate.html, GPCP from https://disc.gsfc.nasa.gov/datasets/GPCPMON_3.1/summary?keywords=GPCPMON, GPM from https://gpm1.gesdisc.eosdis.nasa.gov/data/GPM_L3/, JRA from https://climatedataguide.ucar.edu/climate-data/jra-55 and https://jra.kishou.go.jp/JRA-55/index_en.html, MERRA-2 from https://disc.gsfc.nasa.gov/datasets?project=MERRA-2, NOAA (PREC/LAND) from https://psl.noaa.gov/data/gridded/data.precl.html, PERSIANN (CCS, CDR, CCS-CDR, PDIR-NOW) from https://chrsdata.eng.uci.edu/, TRMM from https://disc.gsfc.nasa.gov/datasets/TRMM_3B43_7/summary, UDEL from https://psl.noaa.gov/data/gridded/data.UDel_AirT_Precip.html. The GCAM model output used in this study is available from https://doi.org/10.25584/data.2020-07.1357/1644253. Source data are provided with this paper.

## Code availability

The code used in this analysis is available through https://doi.org/10.5281/zenodo.7373832.

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

**Acknowledgements** The research has been supported by funding from the European Research Council under the European Union's Horizon 2020 research and innovation programme (DECAF project, grant agreement no. 771492), and the Newton Fund, through the Met Office Climate Science for Service Partnership Brazil.

**Author contributions** All authors developed the concept of the study, and contributed to experimental design, interpretation of results and writing of the manuscript. C.S. and J.C.A.B. contributed to the analysis.

**Competing interests** The authors declare no competing interests.

**Additional information**
**Correspondence and requests for materials** should be addressed to C. Smith.

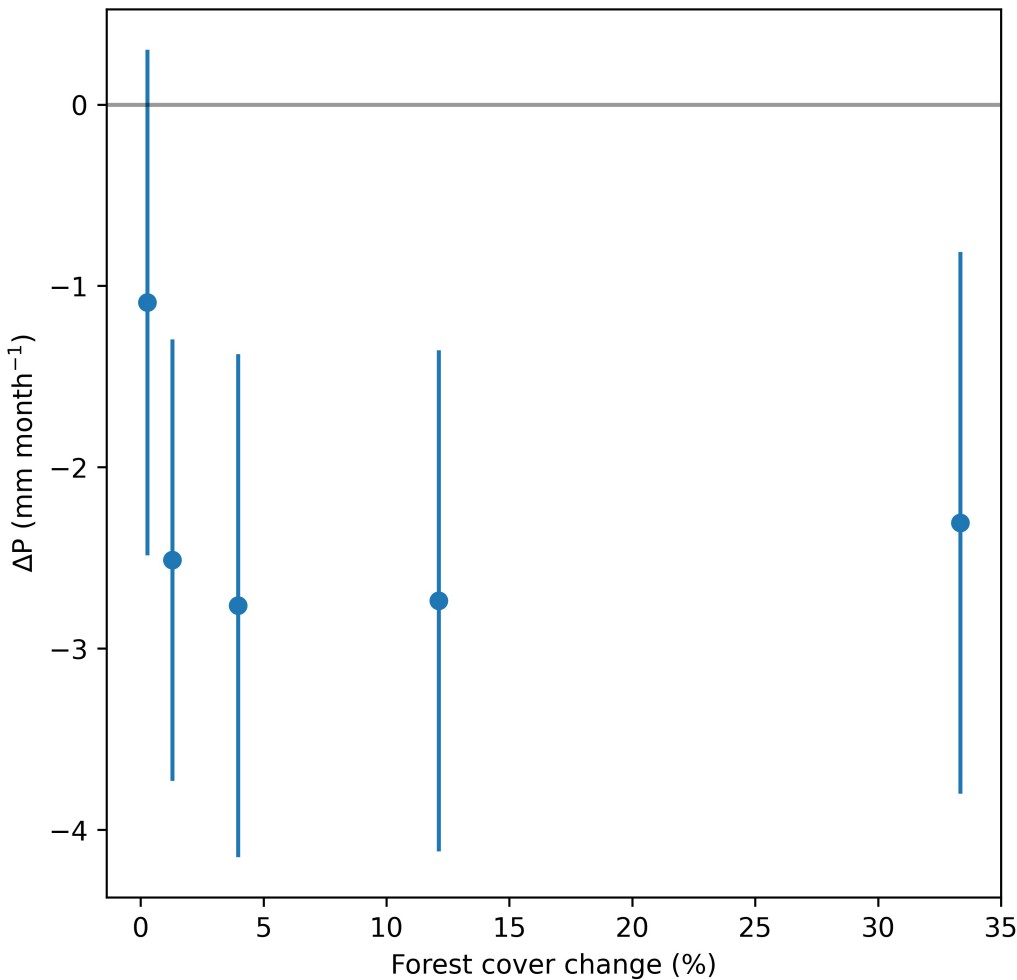

**Extended Data Fig. 1 | Annual precipitation change as a function of forest loss.** Results are shown at 2° spatial resolution for all satellite precipitation (P) datasets calculated as the change in P over time for deforested data pixels minus change over time for control data pixels. Data is binned according to forest cover change (%) with an equal number of pixels in each bin. Points show the median and error bars show ±1 standard error from the mean. Details of each data product are provided in Extended Data Table 1.

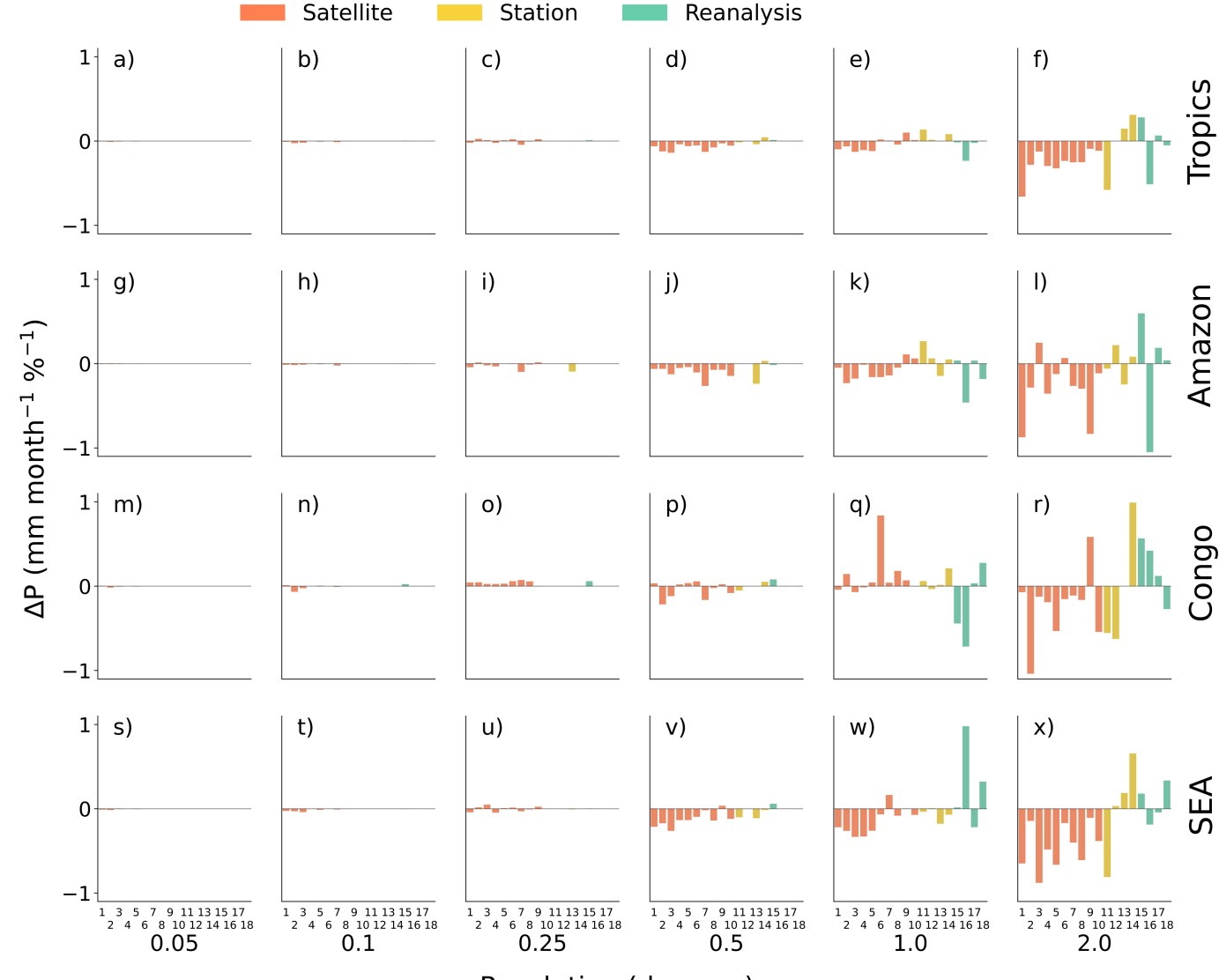

**Extended Data Fig. 2 | Annual precipitation change due to forest loss for individual datasets.** Results are shown for 2003 – 2017 for 5 year averages and 3x3 moving window. Bars show the median absolute change in annual P (mm month⁻¹) per percentage point tree cover loss in each region (Tropics (a-f), Amazon (g-l), Congo (m-r), SEA (s-x)). Each P dataset is shown separately and ordered and coloured by category: satellite (orange), station (yellow) and reanalysis (turquoise). The datasets are numbered; 1) CHIRPS, 2) CMORPH, 3) CPC, 4) CRU, 5) ERA5, 6) GPCC, 7) GPCP, 8) GPM, 9) JRA, 10) MERRA-2, 11) NOAA 12) PERSIANN-CCS, 13) PERSIANN-CCSCDR, 14) PERSIANN-CDR, 15) PERSIANN-NOW, 16) PERSIANN, 17) TRMM, 18) UDEL. Results are shown for forest loss scales of 0.05° (a,g,m,s), 0.1° (b,h,n,t), 0.25° (c,i,o,u), 0.5° (d,j,p,v), 1.0° (e,k,q,w), 2.0° (f,l,r,x). Details of each data product are provided in Extended Data Table 1.

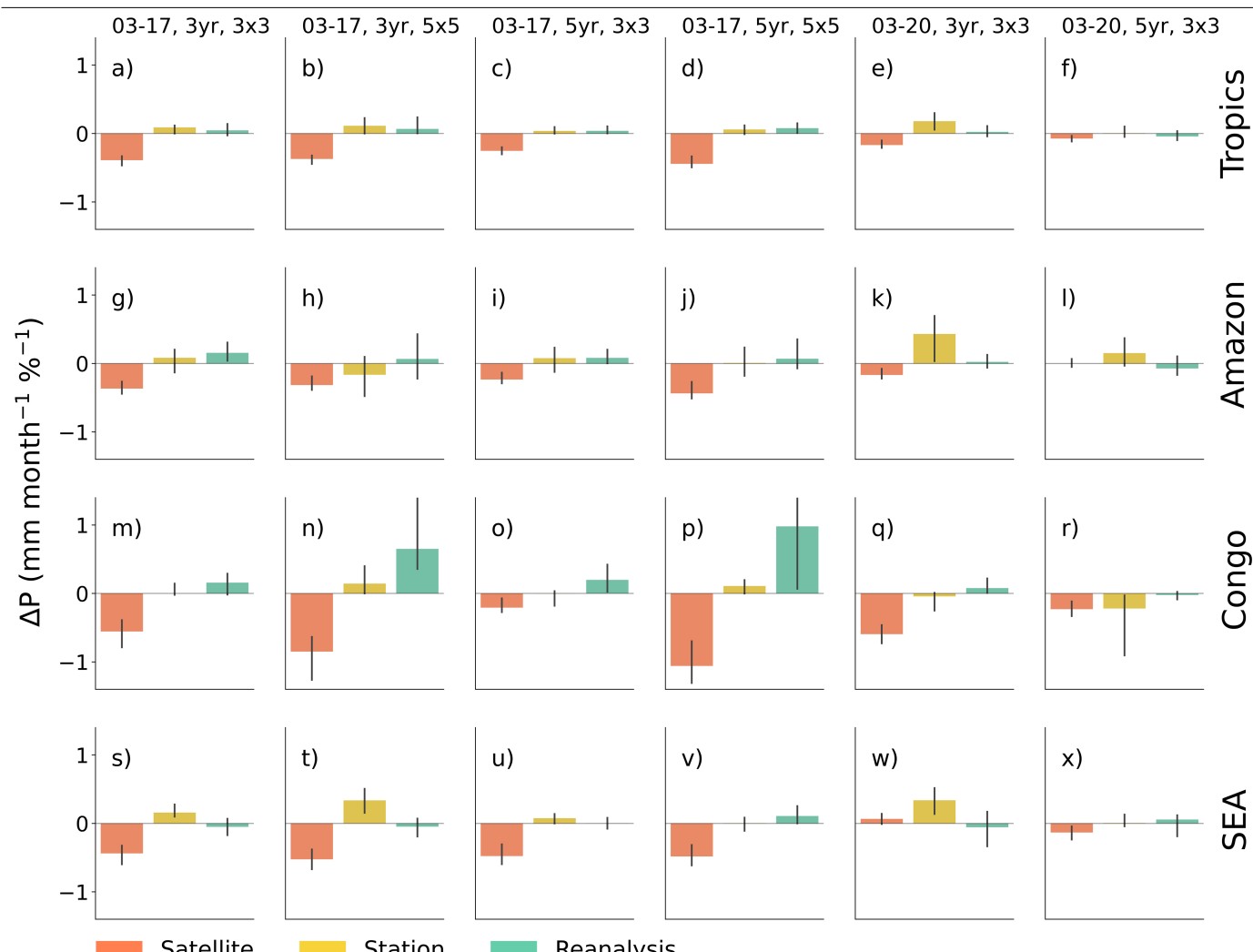

**Extended Data Fig. 3 | Changes in precipitation due to forest loss for different time periods and nearest neighbour comparisons.** Changes in annual mean precipitation at 2.0° resolution are shown for satellite (orange), station (yellow) and reanalysis (turquoise) datasets for the tropics (a-f), Amazon (g-l), Congo (m-r) and Southeast Asia (SEA, s-x). Columns show the sensitivity of our results to changes in the analysis period, number of years used to compute multi-annual means at start and end of the analysis period, and size of the moving window used for nearest neighbour comparisons: 2003-2017, 3-year averages and 3x3 nearest neighbour (Column 1, a,g,m,s); 2003-2017, 3-year, 5x5 (Column 2; b,g,n,t); 2003-2017, 5-year, 3x3 (Column 3; c,i,o,u); 2003-2017, 5-year, 5x5 (Column 4; d,j,p,v); 2003-2020, 3-year, 3x3 (Column 5; e,k,q,w); 2003-2020, 5-year, 3x3 (Column 6; f,l,r,x). Error bars show ± 1 standard error from the mean. Details of each data product are provided in Extended Data Table 1. Full results for all tested resolutions are available in an online repository [10.5281/zenodo.7373832].

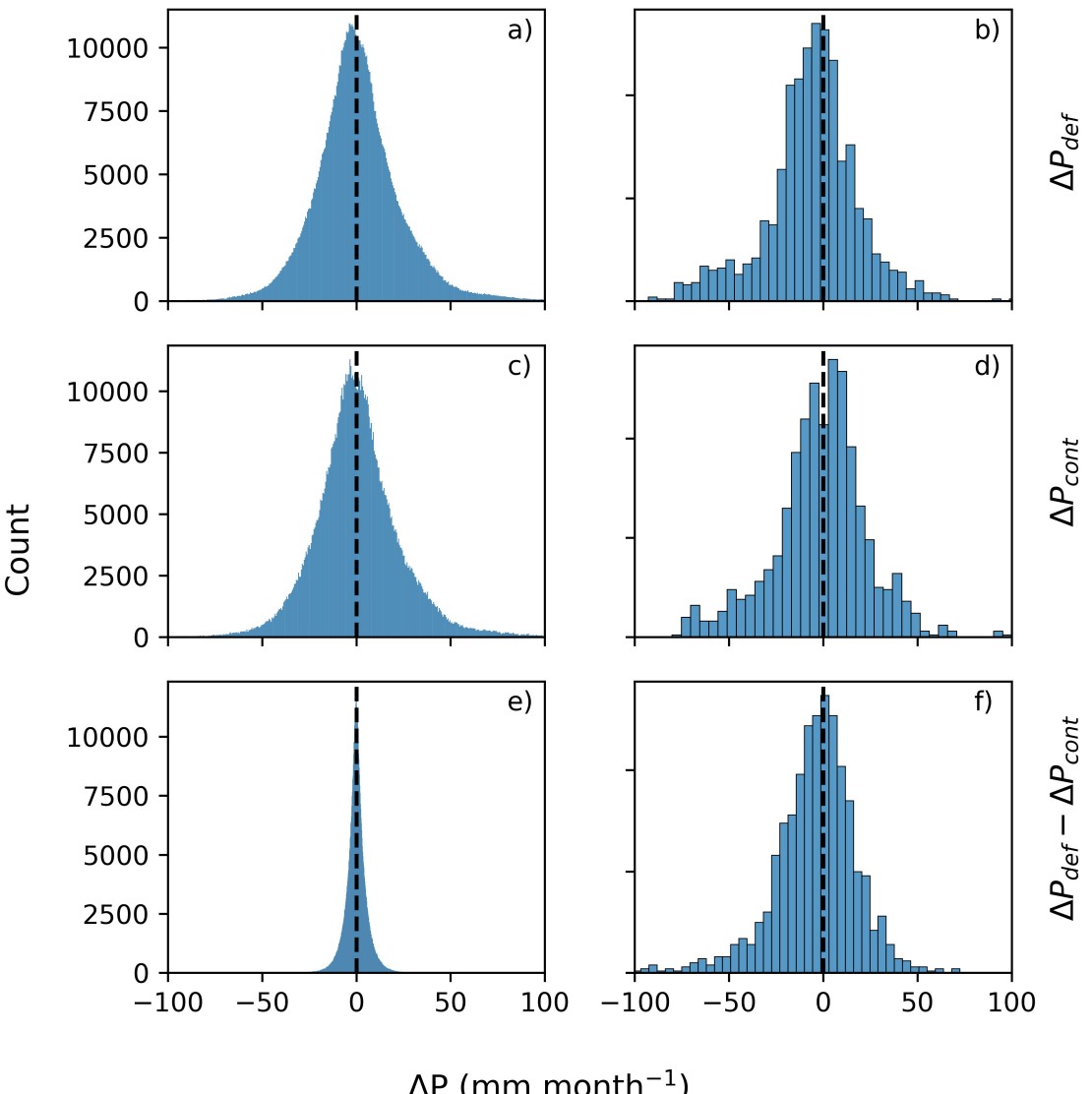

**Extended Data Fig. 4 | Change in precipitation over deforested, control and difference between deforested and control pixels.** Change in precipitation over 2003 to 2017 is shown for deforested (a, b), control (c, d) and difference between deforested and control pixels (e, f) for 0.05° (a, c, e) and 2.0° (b, d, f) resolution. Details of each data product are provided in Extended Data Table 1.

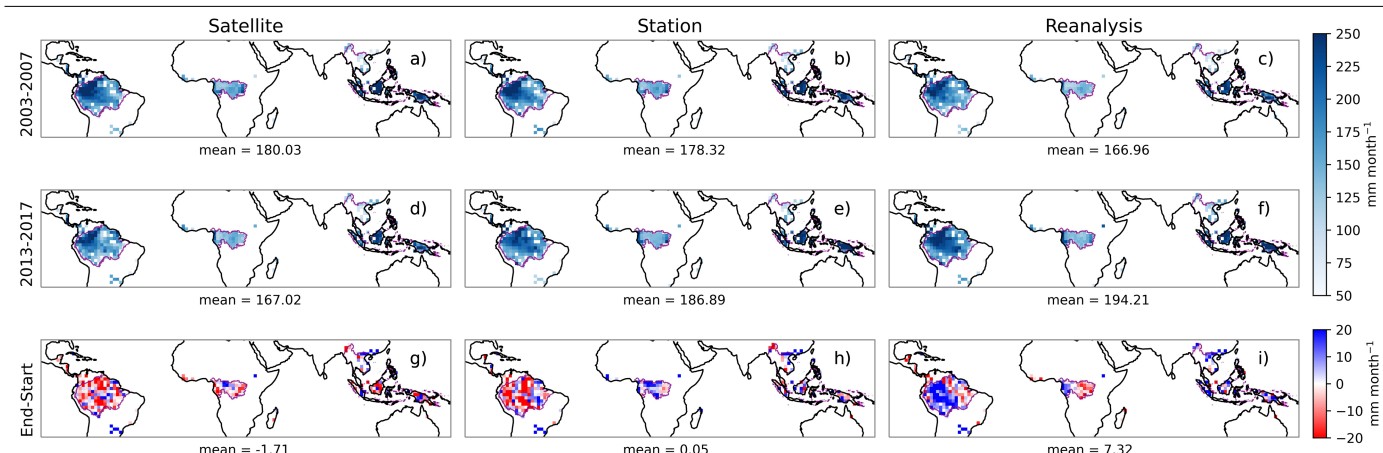

**Extended Data Fig. 5 | Mean precipitation from satellite, station and reanalysis datasets.** For each class of dataset, satellite (a, d, g), station (b, e, h) and reanalysis (c, f, i), the median value for the 5-year multi-annual mean at the start (2003-2007; a, b, c) and end (2013-2017; d, e, f) of the analysis period as well as the change over the analysis period (end – start; g, h, i) is shown. Mean values across tropical evergreen broadleaf forests are shown in units of mm/month at the top of each panel. Maps of the different regions generated using Cartopy and Natural Earth[51]. Details of each data product are provided in Extended Data Table 1.

**Extended Data Table 1 | Precipitation datasets used in this study refs[58-73]**

| Dataset | Date Range | Highest Res | Inputs | Category | Reference |
|---------|-----------|-------------|--------|----------|-----------|
| CHIRPS v2.0 | 1981 - 2021 | 0.05 | Satellite + station | Satellite | Funk et al., 2015 [57] |
| CMORPH | 1998 - 2020 | 0.25 | Satellite + GPCP | Satellite | Xie et al., 2019 [58] |
| CPC | 1979 - 2021 | 0.5 | GTS + COOP | Station | Xie et al., 2007 [59] |
| CRU TS v4.06 | 1901 - 2021 | 0.5 | Station | Station | Harris et al., 2020 [19] |
| ERA5 | 1979 - 2021 | 0.1 | Numerous | Reanalysis | Hersbach et al., 2020 [60] |
| GPCC v2022 | 1891 - 2020 | 0.25 | Stations | Station | Elke et al., 2022 [61] |
| GPCP v3.2 | 1996 -2020 | 0.5 | PERSIANN, GPROF TORVS/AIRS, GPCC GAUGE | Satellite | Huffman et al., 2022 [62] |
| GPM v0.6 | 2000 - 2020 | 0.1 | 9 satellites | Satellite | Hou et al., 2014 [63] |
| JRA v7.0 | 1979 - 2020 | 0.5625 | Numerous | Reanalysis | Kobayashi et al., 2015 [64] |
| MERRA-2 | 1980 - 2020 | 0.625 | Numerous | Reanalysis | Gelaro et al., 2017 [65] |
| NOAA | 1948 - 2021 | 1 | GHCN gauge, CAMS | Reanalysis | Chen et al., 2002 [66] |
| PERSIANN-CCS | 2003 - 2020 | 0.04 | Cloud class system | Satellite | Nguyen et al., 2019 [67] |
| PERSIANN-CDR | 1983 - 2020 | 0.25 | GridSat B1, GPCP | Satellite | Ashouri et al., 2015 [68] |
| PERSIANN-CCSCDR | 2003 - 2020 | 0.04 | GridSat B1, GPCP | Satellite | Sadeghi et al., 2021 [70] |
| PERSIANN_NOW | 2000 - 2020 | 0.04 | IR data, IMERG, WorldClim 2, PERSIANN-CDR, NCEP, GPCP, GSMAP-NOW | Satellite | Nguyen et al., 2020 [69] |
| PERSIANN | 2000 - 2020 | 0.25 | AI on IR and VIS from geostat | Satellite | Nguyen et al., 2019 [67] |
| TRMM v3B43 | 1998 - 2019 | 0.25 | PR, TMI, VIRES CERES, LSI | Satellite | Huffman et al., 2007 [71] |
| UDEL v5.01 | 1990 - 2017 | 0.5 | GHCN, GSOD | Station | Matsuura and Willmott, 2018 [72] |