## [Peer Review File · Nature]

Manuscript Title: Tropical deforestation causes large reductions in observed precipitation

Reviewer Comments & Author Rebuttals

Reviewer Reports on the Initial Version:

Referees' comments:

Referee #1 (Remarks to the Author):

This study by Smith et al. presents important empirical results on the effects of tropical deforestation on precipitation at different spatial scales, up to the scale of 2 degrees (c. 200 km). Full understanding of especially the effects of the spatial scale of deforestation on precipitation is still lacking, so this novel pan-tropical empirical analysis across scales is a very welcome contribution. This work will also prove very useful in steering future research, particularly in the selection of data sources to determine precipitation changes in the tropics. Therefore, I applaud the authors with this study, but I also have some major concerns as follows:

- The 2003-2005 mean precipitation levels are compared to those of 2015-2017. Although I understand that the authors were forced to analyze such short periods due to limited data availability, it means that the effects of climate fluctuations on the decadal scale, such as ENSO, could not be accounted for. If deforestation and effects of such fluctuations coincide (and we know that neither deforestation nor, e.g., ENSO impacts are spatially randomly distributed), it may confound the results.
- Effects of deforestation on precipitation have been estimated by previous studies to greatly exceed the largest spatial scale analyzed here. See Van der Ent & Savenije (2011) for length scales of global atmospheric moisture recycling (500-2000 km for the tropics) and Staal et al. (2018) for length scales of moisture recycling in the Amazon specifically (spatially varying, with a median of 600 km). On the one hand this could mean that the effects of deforestation on precipitation may be underestimated by the current study. On the other hand some estimated effects of local-scale deforestation may be partially influenced by deforestation at regional scales. Although the use of control pixels will generally be statistically correct for the latter effect at the smaller resolutions, it could play a considerable role at especially the 2 degree resolution.

In short, although this work will certainly be publication-worthy, its design also fundamentally prevents it from completely answering the question: what is the relationship between deforestation and precipitation in the tropics?

It is also important to note that magnitude of deforestation beyond those observed in the study period might have consequences that would not be detected by this approach, creating considerable uncertainty in the linearly scaled results presented in Figure 4. Indeed, the papers referenced in lines 149-165 employ different methods and tend to predict larger decreases in precipitation at large-scale deforestation.

Further minor comments:

- Line 24: The Sternberg paper presents a conceptual model, whereas a more empirical reference is more in place here.
- Fig 1: please mention the spatial resolution of the data and their source in the caption.
- Line 74: alpha level of 0.10 is quite high and higher than used elsewhere in the study; why not be consistent?
- The reference in line 536 is a conference abstract, whereas it has also been published as a paper (in Nature Communications).
- The supplementary Figs. 8-10 are very hard to read. Now, essentially the only information that the reader is able to derive from this is that at 2 degrees resolution changes tend to be much greater than at other resolutions.
- The figures in which significance is depicted by the filled versus translucent bars (e.g. Figs. 2, 3 and several in the supplement) are not clear enough. At first sight everything seems significant until one starts to notice the differences in colors that signify significance, which are too subtle in my opinion.

References

- Staal et al. (2018). Forest-rainfall cascades buffer against drought across the Amazon. *Nature Climate Change* 8, 539-543.
- Van der Ent & Savenije (2011). Length and time scales of atmospheric moisture recycling. *Atmospheric Chemistry and Physics* 11, 1853-1863.

Referee #2 (Remarks to the Author):

Numerous studies based on numerical modeling experiments have suggested that large scale deforestation in the Tropics would cause precipitation to decrease, but observational evidence is lacking. Here this manuscript presents an observational study on the impact of tropical deforestation. If confirmed, this would be the first study presenting observational evidence for deforestation-induced precipitation decrease. This topic is of broad interest to Nature's readers and I believe the importance of the finding (if confirmed) is suitable for Nature. I find the results encouraging and also intriguing, and would love to see this published in Nature. However, I have several major concerns that need to be addressed before this manuscript can be considered for publication in any reputable journal:

- 1) Precipitation difference between the first three years and last three years of the period 2003-2017 is used to represent the precipitation changes over this period, and its difference between deforested and chosen control pixels is attributed to deforestation over the same period. Unfortunately, due to the relatively short time period and the strong 2015-2016 El Nino (which caused large negative precipitation anomalies over most of the tropical land), this approach is extremely problematic. El Nino is responsible for much of the "precipitation change" over 2003-2017 derived in this study, which is a huge uncertainty that is not addressed or discussed in this manuscript. Note that deforestation during 2003-2017 would continue to influence the regional climate after 2017 (even if one were to assume no more deforestation took place afterwards), so the period used to quantify precipitation response does not need to end in 2017. Also, if you plot the time series of precipitation during 2003-2022, it will show how representative (or unrepresentative)

the chosen three years are.

2) An important finding is that the deforestation-induced precipitation changes are detectable only under coarse resolution. At the 2 degree resolution, where are the corresponding control pixels? Given the coarse grid, it is highly likely that the control pixels might be in a very different location/regime and therefore the deforestation-control difference is complicated by differences in precipitation variability between them. This is especially the case given the strong El Nino impact and the strong precipitation gradient over most of the region where deforestation occurred. (If this is the case, then the signal would be weaker in coarse resolution station-based or reanalysis data as found in the study because they cannot capture the sharp spatial gradient well). This possibility (that the difference is not related to deforestation) has to be addressed in order to attribute the differences to deforestation.

3) Related to 2), Supplementary Figure 6 shows that as the spatial scale increases (e.g., from 0.5 to 2.0 degrees), the histogram of precipitation changes over deforested pixels shifts from centering around positive values (indicating an increase) towards centering over negative values (indicating a decrease of precipitation). Is such a shift also present over the control pixels? How do similar plots for the control pixels look like? Please present the same analysis over the control pixels. Are the results qualitatively different between the two?

4) Figure 2 shows the spatially averaged precipitation difference per unit loss of forest cover. However, no scatter plot between precipitation difference and forest cover loss at the pixel level is shown. Please show the scatter plot between the two variables. Is there a significant correlation between the two? How does it vary under different resolutions?

5) Figure 4 assumes that the relationship between dP and dC is linear and will continue to be linear. Whether this is linear, and if yes whether the linearity may continue as dC increases, can and must be checked from a scatter plot and regression analysis between the two, because you have the data. This is also related to my comment 4). The scatter plot is essential to the main results yet was not checked (or at least not presented).

6) Figure 3 shows the results for wet season, dry season, and annual precipitation. However, previous studies have shown that in tropical forest regions, precipitation is most sensitive to deforestation or land cover change in transition seasons (e.g., Fu & Li, 2004; Leite-Filho et al., 2019; Jiang et al., 2021). In fact, the results presented in this manuscript showed that the annual precipitation response is much stronger than in both wet and dry seasons in Congo (indicating stronger signal in transition seasons). Please check the other seasons, and discuss your results in the context of existing literature.

Minor:

1) The value of forest cover loss does not change as the pixel size becomes larger (as shown in Supplementary Figure 5). That is counterintuitive. Please double check or otherwise clarify. See for example Jiang et al. (2021).

2) Supplementary figure 11: there seem to be a mistake in the color bars (the color sequence differs between different panels).

3) In several figure captions: "... different to controls ..." should be "different from controls"

References:

Fu & Li, 2004: The influence of the land surface on the transition from dry to wet season in Amazonia. *Theoretical and Applied Climatology*.

Leite-Filho et al., 2019: The southern Amazon rainy season: The role of deforestation and its interactions with large-scale mechanisms. *International Journal of Climatology*.

Jiang et al., 2021: Modeled Response of South American Climate to Three Decades of Deforestation. *Journal of Climate*.

Author Rebuttals to Initial Comments:

Response document for manuscript: "*Tropical deforestation causes large reductions in observed precipitation*", submitted to Nature.

We thank the reviewers for their careful and thoughtful review of our work and for their useful and constructive comments. We thank the editor for the opportunity to respond to these comments. Addressing the reviewer comments has allowed us to demonstrate that our results are robust to a range of different methodological approaches and assumptions. We feel that our revised manuscript clearly demonstrates observable reductions in precipitation over regions of tropical forest loss in the satellite datasets.

In this response document we first provide a summary of the main changes to our work and then provide detailed responses to each of the reviewer comments in turn.

Summary of changes:

ENSO alters precipitation across the tropics and the reviewers highlighted a potential impact of ENSO on our results. We show a consistent and expected decline in precipitation during the 2015/16 period across the datasets used in our analysis. To explore the effect of this on our results we extended our analysis period allowing us to exclude the 2015/16 El Niño period. We show the reductions in precipitation over regions of forest loss persist when we exclude the El Niño years, showing our result is not impacted by ENSO.

The reviewers highlighted our short analysis period and questioned the potential impact of this on our results. To address this, we extended our analysis period (this involved moving to updated datasets in some cases, see below) and lengthened our multi-year mean from 3 to 5 years. We found consistent reductions in precipitation over regions of forest loss using both 3-year and 5-year means at the start and end of our analysis period. For the main paper we changed to using a 5-year mean (rather than 3-year means) at the start and end of the time series.

The reviewers asked us to further explore seasonality and extend our analysis to the transition period between dry season and wet season. We have added this analysis and found deforestation also drives reductions in precipitation during this transition period. Over the Amazon, largest reductions in precipitation occurred in this transition period, in line with previous work.

The reviewers highlighted the need to demonstrate that our predictions of the impacts of projected future deforestation were constrained by our observational analysis. We added analysis to show the sensitivity of precipitation as a function of forest loss, confirming generally greater reductions in precipitation with increased forest loss. To ensure our projections are constrained by observations we have restricted our future projections to the range of forest loss that is robustly sampled in our analysis. Finally, we explored how future projections of precipitation depend on assuming linear or non-linear sensitivity of precipitation to forest loss, confirming important reductions in precipitation under projected deforestation in both approaches.

In addition to the changes outlined below in response to reviewer comments, we have taken this opportunity to update some of the datasets that have newer versions available, these are shown below.

Dataset updates:

- Hansen tree cover data v1.8 -> v1.9
- GPCP v3.1 -> v3.2
- GPCC v2020 -> v2022
- CRU v4.05 -> 4.06
- ERA5 0.25 degrees -> ERA5 0.1 degrees

Below we have addressed each reviewer comment and made references to changed text and figures in both the main paper and the supplementary material. Changes to text have been highlighted in red in the article file and supplementary document.

Referees' comments:

Referee #1 (Remarks to the Author):

Comment 1.1: *This study by Smith et al. presents important empirical results on the effects of tropical deforestation on precipitation at different spatial scales, up to the scale of 2 degrees (c. 200 km). Full understanding of especially the effects of the spatial scale of deforestation on precipitation is still lacking, so this novel pan-tropical empirical analysis across scales is a very welcome contribution. This work will also prove very useful in steering future research, particularly in the selection of data sources to determine precipitation changes in the tropics. Therefore, I applaud the authors with this study, but I also have some major concerns as follows:*

Response 1.1: We thank the reviewer for their work evaluating our manuscript and for their thoughtful suggestions for its improvement.

Comment 1.2: *The 2003-2005 mean precipitation levels are compared to those of 2015-2017. Although I understand that the authors were forced to analyze such short periods due to limited data availability, it means that the effects of climate fluctuations on the decadal scale, such as ENSO, could not be accounted for. If deforestation and effects of such fluctuations coincide (and we know that neither deforestation nor, e.g., ENSO impacts are spatially randomly distributed), it may confound the results.*

Response 1.2: This is an important point, thanks for raising it. We expect that our nearest neighbour approach will largely prevent our results being confounded by ENSO (or other large-scale climate variability), since both deforested and control pixels will be impacted in a similar way by large-scale climate signals. However, we agree that it is important to check that our results were not confounded by the impacts of the strong ENSO event in 2015/16. To do this we followed two approaches: first, we extended the period that we used to calculate multi-annual means at the start and end of the analysis period from three years to five years. This reduces the effect of precipitation anomalies in a single year distorting the results.

Second, by excluding TRMM and UDEL datasets (which are only available until 2019 and 2017 respectively), we extended the analysis to compare mean precipitation levels from 2003-2005 with precipitation in 2018-2020 (Fig. S5), thus excluding the 2015/2016 ENSO completely from the analysis. We also conducted an extended analysis using 5-year means (2003-2005 & 2016-2020, Fig. S4), which excludes the driest ENSO year of 2015. In both of these approaches, we found the precipitation response to deforestation was robustly negative among satellite datasets demonstrating that our reported results are not an artefact of comparing precipitation in non-ENSO years at the start with ENSO years at the end. We found that the precipitation response to forest loss was stronger during the ENSO years, which might be expected given higher transpiration rates observed in tropical forests during ENSO (Brum et al., 2018) and because the impacts of deforestation are greater in drought years.

We add the following text (lines: 138-147):

“Our analysis period includes the 2015-2016 El Niño which resulted in negative precipitation anomalies over many tropical land regions (Fig. S3). We found the precipitation response to forest loss was robustly negative during both El Niño and non- El Niño years (Fig. S4-5). The precipitation response to forest loss was stronger during El Niño years, likely due to higher transpiration rates observed in tropical forests during El Niño years (Brum et al., 2018) and because rainfall is more sensitive to reductions in moisture recycling during drought years (Bagley et al., 2014; Staal et al., 2018). Climate change is expected to lead to increased droughts over many tropical regions (Seneviratne et al., 2021; Wunderling and Staal, 2022), which may be further exacerbated by ongoing deforestation.”

We also add the following text to the Methods (lines: 332-342):

“To explore the role of the analysis period on our results we compared the results for 5-year means to shorter 3-year means (2003-2007 compared to 2015-2017) and found consistent results (Fig. S9). Our analysis period includes the strong 2015/2016 El Niño which resulted in reductions in precipitation over most tropical land regions, particularly in 2015 (Fig. S3). To explore the potential impacts of the 2015/16 El Niño on our analysis we estimated the impact of forest loss on precipitation using 3-year (2003-2005 vs. 2018-2020) and 5-year (2003-2007 vs. 2016-2020) multi-annual means spanning an extended time period. The 3-year analysis completely excludes the 2015/2016 ENSO, whilst the 5-year analysis excludes 2015, which was the driest year (Fig. S4-5). Two datasets (TRMM and UDEL) were not available after 2017 so were removed from this sensitivity analysis.”

Comment 1.3: *Effects of deforestation on precipitation have been estimated by previous studies to greatly exceed the largest spatial scale analyzed here. See Van der Ent & Savenije (2011) for length scales of global atmospheric moisture recycling (500-2000 km for the tropics) and Staal et al. (2018) for length scales of moisture recycling in the Amazon specifically (spatially varying, with a median of 600 km). On the one hand this could mean that the effects of deforestation on precipitation may be underestimated by the current study. On the other hand some estimated effects of local-scale deforestation may be partially influenced by deforestation at regional scales. Although the use of control pixels will generally statistically correct for the latter effect at the smaller resolutions, it could play a considerable role at especially*

the 2 degree resolution.

In short, although this work will certainly be publication-worthy, its design also fundamentally prevents it from completely answering the question: what is the relationship between deforestation and precipitation in the tropics?

Response 1.3: We agree that that limiting our analysis to 2 degrees means that the effect of deforestation on precipitation at scales greater than this could not be quantified. This is a caveat of our analysis approach, since evaluating responses at scales $>2^\circ$ using our moving-window nearest-neighbour methodology would result in too few deforested pixels to draw robust conclusions. Our nearest-neighbour approach, where we compare deforested pixels with neighbouring control pixels, helps to reduce the impact of large-scale climate signals from our analysis. However, as the scale of the analysis increases beyond 2° the comparison between control and deforested pixels becomes more problematic, we have acknowledged these caveats of our approach in the revised version of the paper and make it clear that our precipitation response estimates are missing deforestation impacts at larger spatial scales. We feel that despite these caveats, our analysis makes a valuable contribution to our understanding of how tropical forest loss impacts rainfall across a broad range of scales from 5 km to 200 km.

We add the following text (lines: 243-257):

“Our approach is restricted to observing deforestation impacts up to scales of ~200 km (see Methods). At larger scales, insufficient pixels experienced forest loss during the relatively short period of satellite observations for a robust analysis. Deforestation is also likely to alter rainfall at these larger scales through reducing moisture recycling leading to reductions in rainfall downwind of forest loss (Spracklen et al., 2012; Zemp et al., 2017; Staal et al., 2018; Xu et al., 2022). The length scale of moisture recycling is typically 500 – 2000 km in the tropics (Van Der Ent and Savenije, 2011) with a median value of 600 km in the Amazon (Staal et al., 2018). In regions downwind of extensive forests, such as the southwestern Amazon, up to 70% of precipitation could be sourced from upwind evapotranspiration (van der Ent et al., 2010; Sorí et al., 2017). Tropical forest loss could have severe implications for precipitation in these regions that are 100s - 1000s of km downwind of the forest loss (Staal et al., 2018). Through missing the impacts at these larger scales, our analysis is likely to underestimate the full impacts of deforestation on rainfall.”

Comment 1.4: *It is also important to note that magnitude of deforestation beyond those observed in the study period might have consequences that would not be detected by this approach, creating considerable uncertainty in the linearly scaled results presented in Figure 4. Indeed, the papers referenced in lines 149-165 employ different methods and tend to predict larger decreases in precipitation at large-scale deforestation.*

Response 1.4: This is an important point, thanks for raising it. We agree that our original Fig. 4 calculated precipitation response estimates for canopy loss values higher than the range included in our observational analysis, introducing

considerable uncertainty. We have checked the observed P response to canopy loss and found a robust signal up to 30% canopy loss (see Fig. S7a). For canopy loss greater than 30% there were too few grid cells to draw robust conclusions. For this reason, we have revised Fig. 4 to estimate the precipitation response to canopy losses of up to 30% only. For grid cells where future projected canopy loss exceeds 30%, we cap loss at 30%. We expect our updated figure to provide a conservative estimate of the precipitation response to future forest loss.

We add the following text to explain our approach (lines: 187-196):

“To ensure that our projections are observationally constrained we restrict our analysis to the 0-30% reductions in canopy cover loss that are well sampled in our observational dataset (Fig. S7) and cap the impacts of greater forest loss at that of 30%. Our analysis will underestimate the precipitation impacts over regions projected to experience the most extensive deforestation, including the Congo where mean tree cover is projected to decline by 40 percentage points between 2015 and 2100 (Fig. 4a). We estimate 2015 - 2100 forest loss could lead to reductions of annual mean precipitation of up to 6.5 ± 2.6 mm month⁻¹ in the Congo and 6.2 ± 2.5 mm month⁻¹ in SEA (Fig. 4b), equivalent to precipitation declines of 3-4%.”

We add the following text to acknowledge that our analysis is likely to underestimate the impacts of larger magnitude deforestation (lines: 199-204):

“Our observationally-based approach will miss tipping points in the climate system that might be reached as deforestation extent progresses further. Such tipping points have been postulated for the Amazon under future global change (Lenton et al., 2008; Wunderling and Staal, 2022; Xu et al., 2022). Thus, the substantial declines in precipitation projected in our analysis should be viewed as a conservative estimate of the potential precipitation response to future deforestation.”

Further minor comments:

Comment 1.5: Line 24: The Sternberg paper presents a conceptual model, whereas a more empirical reference is more in place here.

Response 1.5: Resolved

Comment 1.6: Fig 1: please mention the spatial resolution of the data and their source in the caption.

Response 1.6: We have updated the figure caption.

Comment 1.7: Line 74: alpha level of 0.10 is quite high and higher than used elsewhere in the study; why not be consistent?

Response 1.7: We have updated all significance levels to be consistent across the paper.

Comment 1.8: The reference in line 536 is a conference abstract, whereas it has also been published as a paper (in Nature Communications).

Response 1.8: We have updated the reference

Comment 1.9: *The supplementary Figs. 2,20-22 are very hard to read. Now, essentially the only information that the reader is able to derive from this is that at 2 degrees resolution changes tend to be much greater than at other resolutions.*

Response 1.9: We have updated the dataset names along the x axis to numbers with a key which can be read more clearly. This means the reader can better distinguish the behaviour of the individual datasets.

Comment 1.10: *The figures in which significance is depicted by the filled versus translucent bars (e.g. Figs. 2, 3 and several in the supplement) are not clear enough. At first sight everything seems significant until one starts to notice the differences in colors that signify significance, which are too subtle in my opinion.*

Response 1.10: We have updated all our figures to indicate statistical significance using stars to make it much clearer for the reader.

References from reviewer #1:

Staal et al. (2018). Forest-rainfall cascades buffer against drought across the Amazon. *Nature Climate Change* 8, 539-543.

Van der Ent & Savenije (2011). Length and time scales of atmospheric moisture recycling. *Atmospheric Chemistry and Physics* 11, 1853-1863.

Referee #2 (Remarks to the Author):

Comment 2.1: *Numerous studies based on numerical modeling experiments have suggested that large scale deforestation in the Tropics would cause precipitation to decrease, but observational evidence is lacking. Here this manuscript presents an observational study on the impact of tropical deforestation. If confirmed, this would be the first study presenting observational evidence for deforestation-induced precipitation decrease. This topic is of broad interest to Nature's readers and I believe the importance of the finding (if confirmed) is suitable for Nature. I find the results encouraging and also intriguing, and would love to see this published in Nature.*

Response 2.1: We are happy to hear that the reviewer finds our paper interesting and the results important. To address their concerns, we have conducted additional analyses, which further confirm the robustness of our results.

Comment 2.2: *However, I have several major concerns that need to be addressed before this manuscript can be considered for publication in any reputable journal:*
1) *Precipitation difference between the first three years and last three years of the period 2003-2017 is used to represent the precipitation changes over this period, and its difference between deforested and chosen control pixels is attributed to deforestation over the same period. Unfortunately, due to the relatively short time period and the strong 2015-2016 El Nino (which caused large negative precipitation*

anomalies over most of the tropical land), this approach is extremely problematic. El Nino is responsible for much of the “precipitation change” over 2003-2017 derived in this study, which is a huge uncertainty that is not addressed or discussed in this manuscript. Note that deforestation during 2003-2017 would continue to influence the regional climate after 2017 (even if one were to assume no more deforestation took place afterwards), so the period used to quantify precipitation response does not need to end in 2017. Also, if you plot the time series of precipitation during 2003-2022, it will show how representative (or unrepresentative) the chosen three years are.

Response 2.2: We acknowledge the reviewer’s valid concern that our results may be confounded by the strong ENSO event in 2015/16, which caused strong negative precipitation anomalies across the tropics. This concern was also raised by the first reviewer. We refer the reviewer to Response 1.2, which details sensitivity tests performed to check the robustness of the results. We note that our moving window approach should reduce any potential impacts of El Nino or other large scale climate variability through comparing deforestation pixels with neighbouring control pixels over the same time period. As suggested by the reviewer we have extended the time series beyond 2017 and show a robust response a) when comparing longer 5 year means (where the impact of the 2015/2016 will be weaker) and b) when we exclude the 2015/2016 El Nino years from the analysis. This analysis confirms that we observe reductions in precipitation over regions of deforestation even when El Nino years are excluded from the analysis.

We have added a timeseries of the precipitation to the supplementary material (Fig. S3) as suggested. As expected, this shows the reduction in precipitation during 2015/2016, with the largest reduction in 2015 and precipitation returning to normal in 2016/7.

Comment 2.3: *2) An important finding is that the deforestation-induced precipitation changes are detectable only under coarse resolution. At the 2 degree resolution, where are the corresponding control pixels? Given the coarse grid, it is highly likely that the control pixels might be in a very different location/regime and therefore the deforestation-control difference is complicated by differences in precipitation variability between them. This is especially the case given the strong El Nino impact and the strong precipitation gradient over most of the region where deforestation occurred. (If this is the case, then the signal would be weaker in coarse resolution station-based or reanalysis data as found in the study because they cannot capture the sharp spatial gradient well). This possibility (that the difference is not related to deforestation) has to be addressed in order to attribute the differences to deforestation.*

Response 2.3: Due to our nearest-neighbour approach, as the horizontal resolution of the analysis increased, the distance between deforested and control pixels necessarily increased. This could mean that the background climate of control and deforested pixels at 2 degrees is likely to be less similar than at finer resolutions. To

ensure that potential differences in background climate do not influence our results we added an additional analysis where we only consider pixels where climatological monthly mean precipitation differed by less than 10%. We found consistent results with our previous analysis (see Fig S12), indicating that the reported precipitation differences are attributable to differences in canopy loss. We selected a 10% difference based on an analysis of the precipitation distribution of control and deforested pixels (see Fig. S17-18). Furthermore, in response to comment 2.2 we added analysis to remove El Nino years and found consistent reductions in precipitation over regions of deforestation. Together we hope that this additional analysis provides strong confidence that our results are due to deforestation.

We add the following text to the Methods (lines: 332-338):

“To ensure that control pixels and deforested pixels experience a similar background climate we conducted a sensitivity test where we restricted our analysis to pixels where the pre-deforestation precipitation across the control and deforested pixels differed by less than 10%. Restricting our analysis in this way had little impact on our results (Fig. S12) showing that our nearest neighbour approach is effective even at the largest scales analysed here.”

Comment 2.4: *3) Related to 2), Supplementary Figure 17 shows that as the spatial scale increases (e.g., from 0.5 to 2.0 degrees), the histogram of precipitation changes over deforested pixels shifts from centering around positive values (indicating an increase) towards centering over negative values (indicating a decrease of precipitation). Is such a shift also present over the control pixels? How do similar plots for the control pixels look like? Please present the same analysis over the control pixels. Are the results qualitatively different between the two?*

Response 2.4: Thanks for spotting this. We have found that there was a problem with our plot in the original figure. We have corrected this mistake and the new updated figure can be seen in Fig. S17. As suggested, we have added a plot showing the P changes over control pixels can be seen in Fig. S18. The two plots show similarly normally distributed data over both control and deforested pixels.

Comment 2.5: *4) Figure 2 shows the spatially averaged precipitation difference per unit loss of forest cover. However, no scatter plot between precipitation difference and forest cover loss at the pixel level is shown. Please show the scatter plot between the two variables. Is there a significant correlation between the two? How does it vary under different resolutions?*

Response 2.5: This is an important point, and we were remiss not to have presented this analysis in our original submission. We have added a scatter plot showing the relationship between the satellite precipitation response and forest cover loss at 2-degree resolution (Fig. S1). In this analysis we bin the precipitation data into five forest loss bins with an equal number of data points within each bin. We only make predictions of the impact of projected deforestation at the 2 degree resolution, so we restrict our analysis to this scale. This analysis confirms that there

are generally larger reductions in precipitation over areas with greater forest cover loss, though there is considerable variability. Indeed, this plot looks similar to the modelled response of precipitation change as a function of forest loss such as shown in Jiang et al. (2021). We use the uncertainty in the sensitivity of precipitation to forest loss (see error bars in Fig. 2) to provide an uncertainty range in our predictions of precipitation change to future forest loss (see response 2.6 below). We feel this uncertainty analysis helps to better reflect the uncertainty in our projections.

We note this issue is only relevant to the final aspect of our analysis where we make predictions of the impacts of future deforestation on precipitation.

We add the following text (lines: 104-107):

“Analysis of precipitation change as a function of forest loss confirms larger reductions in precipitation for larger reductions in forest cover (Fig. S1), although with considerable variability, as seen in the modelled response (Jiang et al., 2021).”

We now account for this uncertainty in our projections in Fig 4, see response 2.6.

Comment 2.6: 5) Figure 4 assumes that the relationship between dP and dC is linear and will continue to be linear. Whether this is linear, and if yes whether the linearity may continue as dC increases, can and must be checked from a scatter plot and regression analysis between the two, because you have the data. This is also related to my comment 4). The scatter plot is essential to the main results yet was not checked (or at least not presented).

Response 2.6: Following comment 2.5, we have added a plot (Fig. S1) showing the precipitation response to deforestation as a function of forest loss at 2-degrees resolution (the resolution we use to make projections of the impacts of future forest loss). As discussed above this confirms that the precipitation response increases as canopy cover loss increases, although as acknowledged about there is considerable variability.

The referee is correct that we should not assume the precipitation response becomes larger beyond reductions in canopy cover observed in our analysis. To ensure that we do not extrapolate the impacts of future deforestation beyond the range in our observed analysis, we cap forest loss at 30% canopy cover loss, meaning we restrict our analysis to the 0-30% forest loss that is well sampled in our analysis (see response to reviewer to Response 1.4). We have also added a plot showing the distribution of canopy cover loss in the observations and in the future projections (Fig. S7), further showing that a reasonable range for our projections is 0-30% canopy cover loss.

Figure 4 now shows these updated results which moderate our predicted reductions in precipitation due to projected forest loss (since we now cap predictions at canopy loss of 30%). We now predict reductions in precipitation of up to 6 mm month⁻¹ over the Congo and southeast Asia. To better acknowledge the uncertainty in our future

projections we also estimate an uncertainty range based on the uncertainty in our $\Delta P/\Delta CC$ values (represented by the error bars in Fig. 2) and apply these to our projections. This uncertainty is shown as the shading in Figure 4.

We add the following text (lines: 367-372):

“To ensure our results are constrained by our observational analysis we capped future forest loss in each pixel at 30%, which is the upper range of forest loss that is well sampled in the observations (Fig. S13, S1, S7). To estimate the uncertainty in our predictions we applied an upper and lower limit on the sensitivity of precipitation to forest loss based on the median value \pm the standard error (see error bars in Fig. 2) and rescale by forest loss. This provides a range of estimated precipitation impacts of future forest loss.”

Finally, we conduct a sensitivity study where we test the impact of assuming a linear response of future precipitation change to canopy cover loss (as assumed in Fig. 4). We fit a non-linear function through the data presented in Fig. S1 and scale the $\Delta P/\Delta CC$ from this curve by the projected canopy cover loss. This reduces our projected precipitation response (see Fig. S23 and the text below), although the reductions are still considerable.

We add the following text (lines: 383-388):

“We tested the sensitivity of assuming a linear response of precipitation to canopy cover loss. We fitted a non-linear function to the data presented in Fig. S1 and scale the sensitivity of precipitation to forest cover loss ($\text{mm month}^{-1} / \%$) from this function by the projected forest cover loss. This approach (Fig. S23) reduces the projected reduction in precipitation to 3 mm month^{-1} in SEA and $2.5 \text{ mm month}^{-1}$ in the Congo.”

We hope the combination of this new analysis, our modified approach to ensure we do not extrapolate the impacts of forest loss that are outside our observational constraints, and an assessment of the uncertainty provides a better and more robust projection. We agree that it is important to acknowledge the inherent uncertainties in our analysis and we hope our revised manuscript does this more clearly. We note that these uncertainties are likely to reduce in the future as data records extend providing longer timeseries of deforestation and rainfall. This will help to further constrain the impacts of higher deforestation extents, that the reviewers suggest we may be underestimating in our analysis. However, we think that it is important to present our best estimate of the potential impacts of future forest loss on rainfall, whilst acknowledging the uncertainties and clarifying that our estimates are likely to underestimate the actual response.

Comment 2.7: *6) Figure 3 shows the results for wet season, dry season, and annual precipitation. However, previous studies have shown that in tropical forest regions, precipitation is most sensitive to deforestation or land cover change in transition seasons (e.g., Fu & Li, 2004; Leite-Filho et al., 2019; Jiang et al., 2021). In fact, the results presented in this manuscript showed that the annual precipitation response is much stronger than in both wet and dry seasons in Congo (indicating stronger signal*

in transition seasons). Please check the other seasons, and discuss your results in the context of existing literature.

Response 2.7: This is an excellent suggestion, thank you. We have extended our analysis to consider the precipitation response in the transition months, between the driest and the wettest parts of the year. This is now shown in Figure 3. The results confirm deforestation also causes reductions in precipitation in the transition seasons. As anticipated by the reviewer, we find deforestation causes the largest reductions in precipitation during the transition period in the Amazon.

We add the following text (lines: 152-155):

“In the Amazon, deforestation causes the largest reductions in precipitation during the transition season (Fig. 3b) as has been found previously (Fu and Li, 2004; Leite-Filho et al., 2019; Jiang et al., 2021).”

Minor:

Comment 2.8: 1) *The value of forest cover loss does not change as the pixel size becomes larger (as shown in Supplementary Figure 19). That is counterintuitive. Please double check or otherwise clarify. See for example Jiang et al. (2021).*

Response 2.8: Thanks for raising this issue. Fig. S19 does not show forest loss, but instead shows forest loss of the deforested pixel compared to forest loss of the neighbouring control pixel. We have now revised the caption to make this clear. This explains why the values do not change in an intuitive way as they do in the cited paper.

Comment 2.9: 2) *Supplementary figure 11: there seem to be a mistake in the color bars (the color sequence differs between different panels).*

Response 2.9: Removed and otherwise corrected elsewhere.

Comment 2.9: 3) *In several figure captions: “.... different to controls ...” should be “different from controls”*

Response 2.10: Corrected.

References from reviewer #2:

Fu & Li, 2004: The influence of the land surface on the transition from dry to wet season in Amazonia. Theoretical and Applied Climatology.

Leite-Filho et al., 2019: The southern Amazon rainy season: The role of deforestation and its interactions with large-scale mechanisms. International Journal of Climatology.

Jiang et al., 2021: Modeled Response of South American Climate to Three Decades of Deforestation. Journal of Climate.

Response document references:

Bagley, J.E., Desai, A.R., Harding, K.J., Snyder, P.K. and Foley, J.A. 2014. Drought and deforestation: Has land cover change influenced recent precipitation

- extremes in the Amazon? *Journal of Climate*. **27**(1), pp.345–361.
- Brum, M., López, J.G., Asbjornsen, H., Licata, J., Pypker, T., Sanchez, G. and Oiveira, R.S. 2018. ENSO effects on the transpiration of eastern Amazon trees. *Philosophical Transactions of the Royal Society B: Biological Sciences*. **373**(1760).
- Van Der Ent, R.J. and Savenije, H.H.G. 2011. Length and time scales of atmospheric moisture recycling. *Atmospheric Chemistry and Physics*. **11**(5), pp.1853–1863.
- van der Ent, R.J., Savenije, H.H.G., Schaefli, B. and Steele-Dunne, S.C. 2010. Origin and fate of atmospheric moisture over continents. *Water Resources Research*. **46**(9), pp.1–12.
- Jiang, Y., Wang, G., Liu, W., Erfanian, A., Peng, Q. and Fu, R. 2021. Modeled response of south american climate to three decades of deforestation. *Journal of Climate*. **34**(6), pp.2189–2203.
- Lenton, T.M., Held, H., Kriegler, E., Hall, J.W., Lucht, W., Rahmstorf, S. and Schellnhuber, H.J. 2008. Tipping elements in the Earth's climate system. *Proceedings of the National Academy of Sciences of the United States of America*. **105**(6), pp.1786–1793.
- Seneviratne, S.I., Xuebin Zhang, Muhammad Adnan, Mafae Badi, Claudine Dereczynski, Alejandro Di Luca, Subimal Ghosh, Iskhaq Iskandar, James Kossin, Sophie Lewis, Friederike Otto, Izidine Pinto, Masaki Satoh, Sergio M. Vicente-Serrano, Michael Wehner and Botao Zhou 2021. *Weather and Climate Extreme Events in a Changing Climate Coordinating Lead Authors*.
- Sorí, R., Nieto, R., Vicente-Serrano, S.M., Drumond, A. and Gimeno, L. 2017. A Lagrangian perspective of the hydrological cycle in the Congo River basin. *Earth System Dynamics*. **8**(3), pp.653–675.
- Spracklen, D. V., Arnold, S.R. and Taylor, C.M. 2012. Observations of increased tropical rainfall preceded by air passage over forests. *Nature*. **489**(7415), pp.282–285.
- Staal, A., Tuinenburg, O.A., Bosmans, J.H.C., Holmgren, M., Van Nes, E.H., Scheffer, M., Zemp, D.C. and Dekker, S.C. 2018. Forest-rainfall cascades buffer against drought across the Amazon. *Nature Climate Change*. **8**(6), pp.539–543.
- Wunderling, N. and Staal, A. 2022. Recurrent droughts increase risk of cascading tipping events by outpacing adaptive capacities in the Amazon rainforest. , pp.1–11.
- Xu, X., Zhang, X., Riley, W.J., Xue, Y., Nobre, C.A., Lovejoy, T.E. and Jia, G. 2022. Deforestation triggering irreversible transition in Amazon hydrological cycle. *Environmental Research Letters*. **17**(3), p.034037.
- Zemp, D.C., Schleussner, C.F., Barbosa, H.M.J., Hirota, M., Montade, V., Sampaio, G., Staal, A., Wang-Erlandsson, L. and Rammig, A. 2017. Self-amplified Amazon forest loss due to vegetation-atmosphere feedbacks. *Nature Communications*. **8**, pp.1–10.

Reviewer Reports on the First Revision:

Referees' comments:

Referee #1 (Remarks to the Author):

I appreciate the additional analyses by the authors, which further support the robustness of the results. I am also happy that the omission of rainfall effects of deforestation on scales larger than 2 degrees is now better acknowledged. While complimenting the authors on the improved manuscript, I do have a number of final remarks.

I understand the reasoning behind capping all rainfall projections due to deforestation at 30% deforestation, but there is no physical reason why the effects would stop at the currently maximum observed deforestation at the 2 degree scale. Therefore, I believe that extrapolation would be justified, although a sensitivity analysis for different assumptions would be needed in that case. Right now, the estimated future effects of deforestation on rainfall will almost certainly be underestimated, limiting the usefulness and relevance of the respective results.

I appreciate the nonlinear fit as mentioned in lines 374-379, but details on the nonlinear function are missing. As it stands, it cannot be reproduced or interpreted well, so this needs to be improved.

Just a suggestion, but will the results be downloadable also? That is, will the authors provide maps of historical rainfall changes due to deforestation to be used by others?

The abstract notes in line 15 that the greatest reduction in precipitation is observed at 200 km, but it is not clear from the abstract that this is the largest scale that was studied. This may cause readers to assume that precipitation reduction declines at larger spatial scales, for which there is no evidence.

In line 17 it is stated that reanalysis and station-based products “do not capture observed precipitation responses”, but this statement seems to me as being too generalized. For instance, looking at supplementary figure 4, I would say that dataset 11 is very consistent with the satellite products. Also, one should keep in mind that the satellite products are not perfect either, so deviation from satellite-based estimates are not wrong by definition.

Further minor points:

- In Figs. 2 & 3: Given that the dimension is $\Delta P/\%$, the unit would be (mm / month / %), so the closing brackets should be put outside the percentage sign.
- 142 & 198 Ref Wunderling & Staal (2022) should be Wunderling et al. (2022).
- 183 A reference for the high-deforestation scenario is lacking.
- 243 Suggested rewrite: “is typically” -> “has been estimated at” given that their method to get to these numbers is rather indirect.
- In the caption of supplementary figure 2, it should be “ERA5”, as the 5 is not a version number but part of the name. Possibly this applies to MERRA2 as well.
- Please check the reference list very carefully. Many references seem to be missing essential information: often they miss article numbers, but sometimes also other bibliographic info. This

seems to be the case for at least the references in lines 519, 522, 525, 528, 532, 571, 619, 642, 649, 653, 684, 699, 701 and 707.

Referee #2 (Remarks to the Author):

I commend the authors for having addressed most of my early comments with additional analysis. Their approach toward addressing comments from other reviewers seems reasonable to me too. I am somewhat convinced that the differences in precipitation change between deforested and control areas do indicate an impact of deforestation. However, I still have a couple of concerns about the presentation especially about how uncertainties (reflected by the new results) are described or treated:

1) There is a large degree of uncertainty in the relationship between the magnitude of precipitation decrease and the magnitude of deforestation, as reflect in Figure S1. It is puzzling that the precipitation effects seemed to jump to a certain magnitude at a fairly low fraction of forest loss and then plateaued. The manuscript did not explicitly describe this result other than vaguely mentioning the presence of uncertainty. Due to the importance of this particular result and its high relevance to future projection conducted in this study, it should be explicitly discussed (where the figure was referred to or at the end of the manuscript). This will help readers put the main finding of the paper in perspective. It will be also helpful for the authors to comment on what might be the cause for such a response (other than simply the large degree of uncertainty).

2) In response to both reviewers' comments on the need to eliminate the impact of the strong El Nino event at the end of the time series, the authors conducted sensitivity analysis including extending the data to 2020. However, results presented in the main text is still based on the record up to 2017 and therefore includes the impact of the strong El Nino event. Compared to results based on the record up to 2020, the signal for some regions are quite different (in magnitude and in some cases in the direction of changes). Results from the sensitivity analysis are presented in the supplementary figures, and only mentioned in the Method section and again without indicating the specific differences. Given the availability of data up to 2020, readers will likely consider presenting the 2017 results in the main text as cherry picking. One can still present the 2017 results in the main text, only if an explicit description of the discrepancies is provided in the main text Results section (as opposed to the Method section).

Author Rebuttals to First Revision:

Response to Review document for manuscript: "*Tropical deforestation causes large reductions in observed precipitation*", submitted to Nature.

We would like to thank the editor and the two reviewers for their support and insight in further improving our manuscript. We have responded to all the suggestions and comments and have revised our manuscript accordingly. We respond in turn to all the comments below. Where text has been added to the manuscript, line numbers have been referenced and additional text appears in blue in the updated manuscript.

We appreciate the guidance on synthesising results from our sensitivity analysis that was shown in our Supplementary Figures. As suggested, we have now merged many of the supplementary figures into new summary figures in the Extended Data. These new summary figures nicely synthesise the key results of the sensitivity analysis and help with comprehension of our results. We have prepared the datasets and analysis code for upload to a suitable data repository. We hope that our amendments and responses to the comments are acceptable.

Reviewer #1 comments:

Comment 2.1: I appreciate the additional analyses by the authors, which further support the robustness of the results. I am also happy that the omission of rainfall effects of deforestation on scales larger than 2 degrees is now better acknowledged. While complimenting the authors on the improved manuscript, I do have a number of final remarks.

I understand the reasoning behind capping all rainfall projections due to deforestation at 30% deforestation, but there is no physical reason why the effects would stop at the currently maximum observed deforestation at the 2 degree scale. Therefore, I believe that extrapolation would be justified, although a sensitivity analysis for different assumptions would be needed in that case. Right now, the estimated future effects of deforestation on rainfall will almost certainly be underestimated, limiting the usefulness and relevance of the respective results.

Response 2.1: Thanks for this suggestion. We now present results from the uncapped analysis in the main paper (new Fig. 4). In the text of the paper we discuss these results and compare against the results when we cap forest loss at 30% (Fig. S5). We add a short discussion highlighting the larger uncertainty of the precipitation response to forest loss for reductions in forest loss greater than 30% (lines 157-165).

Comment 2.2: I appreciate the nonlinear fit as mentioned in lines 374-379, but details on the nonlinear function are missing. As it stands, it cannot be reproduced or interpreted well, so this needs to be improved.

Response 2.2: Thanks for spotting this. We fit a non-linear function to the data in Extended Data Fig 1. For each forest cover loss bin we apply the median sensitivity of precipitation to forest loss. We have added information to the methods describing our approach (lines 352-354).

Comment 2.3: Just a suggestion, but will the results be downloadable also? That is, will the authors provide maps of historical rainfall changes due to deforestation to be used by others?

Response 2.3: We are keen to provide our data for use by others. The data and code for the paper will be made available online through a public data repository. Associated DOIs will be provided upon acceptance (see Data Availability statement).

Comment 2.4: The abstract notes in line 15 that the greatest reduction in precipitation is observed at 200 km, but it is not clear from the abstract that this is the largest scale that was studied. This may cause readers to assume that precipitation reduction declines at larger spatial scales, for which there is no evidence.

Response 2.4: Text added to line 15: “the largest scale explored in our analysis”.

Comment 2.5: In line 17 it is stated that reanalysis and station-based products “do not capture observed precipitation responses”, but this statement seems to me as being too generalized. For instance, looking at supplementary figure 4, I would say that dataset 11 is very consistent with the satellite products. Also, one should keep in mind that the satellite products are not perfect either, so deviation from satellite-based estimates are not wrong by definition.

Response 2.4: Text changed to: “Reanalysis and station-based products disagree on the direction of precipitation responses to forest loss”. We agree that satellite-based products also contain uncertainty. We cite previous work comparing and discussing different precipitation datasets over the Amazon (Fassoni-Andrade et al. 2021)

Comment 2.6: In Figs. 2 & 3: Given that the dimension is $\Delta P/\%$, the unit would be (mm / month / %), so the closing brackets should be put outside the percentage sign.

Response 2.6: Thank you for pointing this out, we have now changed throughout.

Comment 2.7: 142 & 198 Ref Wunderling & Staal (2022) should be Wunderling et al. (2022).

Response 2.7: Reference amended.

Comment 2.8: 183 A reference for the high-deforestation scenario is lacking.

Response 2.8: We have added a reference to Chen et al., (2020).

Chen, M. et al. Global land use for 2015–2100 at 0.05° resolution under diverse socioeconomic and climate scenarios. *Sci. Data* 7, 1–11 (2020).

Comment 2.9: 243 Suggested rewrite: "is typically" -> "has been estimated at" given that their method to get to these numbers is rather indirect.

Response 2.9: Suggestion accepted, text now changed to "has been estimated at".

Comment 2.10: In the caption of supplementary figure 2, it should be "ERA5", as the 5 is not a version number but part of the name. Possibly this applies to MERRA2 as well.

Response 2.10: Thank you for pointing this out, changes have been made throughout.

Comment 2.11: Please check the reference list very carefully. Many references seem to be missing essential information: often they miss article numbers, but sometimes also other bibliographic info. This seems to be the case for at least the references in lines 519, 522, 525, 528, 532, 571, 619, 642, 649, 653, 684, 699, 701 and 707.

Response 2.11: References have been checked and reformatted to the Nature style.

Reviewer #2 comments:

Comment 3.1: I commend the authors for having addressed most of my early comments with additional analysis. Their approach toward addressing comments from other reviewers seems reasonable to me too. I am somewhat convinced that the differences in precipitation change between deforested and control areas do indicate an impact of deforestation. However, I still have a couple of concerns about the presentation especially about how uncertainties (reflected by the new results) are described or treated:

1) There is a large degree of uncertainty in the relationship between the magnitude of precipitation decrease and the magnitude of deforestation, as reflect in Figure S1. It

is puzzling that the precipitation effects seemed to jump to a certain magnitude at a fairly low fraction of forest loss and then plateaued. The manuscript did not explicitly describe this result other than vaguely mentioning the presence of uncertainty. Due to the importance of this particular result and its high relevance to future projection conducted in this study, it should be explicitly discussed (where the figure was referred to or at the end of the manuscript). This will help readers put the main finding of the paper in perspective. It will be also helpful for the authors to comment on what might be the cause for such a response (other than simply the large degree of uncertainty).

Response 3.1: As suggested we have added a short section towards the end of the manuscript where we discuss uncertainties in our projections of precipitation response to future forest loss (lines 157-165). This uncertainty impacts our precipitation projections presented in Figure 4. We present and discuss the results from our different approaches (capped forest loss at 30%, non-linear response of precipitation to forest loss). We agree that this helps put the projected precipitation changes into perspective.

Comment 3.2: 2) In response to both reviewers' comments on the need to eliminate the impact of the strong El Nino event at the end of the time series, the authors conducted sensitivity analysis including extending the data to 2020. However, results presented in the main text is still based on the record up to 2017 and therefore includes the impact of the strong El Nino event. Compared to results based on the record up to 2020, the signal for some regions are quite different (in magnitude and in some cases in the direction of changes). Results from the sensitivity analysis are presented in the supplementary figures, and only mentioned in the Method section and again without indicating the specific differences. Given the availability of data up to 2020, readers will likely consider presenting the 2017 results in the main text as cherry picking. One can still present the 2017 results in the main text, only if an explicit description of the discrepancies is provided in the main text Results

section (as opposed to the Method section).

Response 3.2: We now provide an explicit description of the results of our sensitivity analyses in the main text (lines 100-108). Specifically, we focus on the regional differences with stronger reductions in precipitation over regions of forest loss during El Nino observed in the Amazon and SEA but not in the Congo. This is likely due to regional differences in the impacts of El Nino, with smaller impacts in the Congo.

Reviewer Reports on the Second Revision:

Referees' comments:

Referee #1 (Remarks to the Author):

I am happy that the authors have incorporated my final comments. Only the article numbers have not been added to the references, but this can be fixed in the proofreading stage. Congratulations on this important and interesting paper.

Referee #2 (Remarks to the Author):

The revision and re-organization of the results look reasonable to me. There is however a major mistake in the Figure ED 3. The 20-year period results (the last two columns) are exactly the same; the results for 5-year averages (the one showing the most sensitivity) are missing!